# Primal-Dual Neural Algorithmic Reasoning

Yu He [1]    Ellen Vitercik [1] [2]

## Abstract

Neural Algorithmic Reasoning (NAR) trains neural networks to simulate classical algorithms, enabling structured and interpretable reasoning over complex data. While prior research has predominantly focused on learning exact algorithms for polynomial-time-solvable problems, extending NAR to harder problems remains an open challenge. In this work, we introduce a general NAR framework grounded in the primal-dual paradigm, a classical method for designing efficient approximation algorithms. By leveraging a bipartite representation between primal and dual variables, we establish an alignment between primal-dual algorithms and Graph Neural Networks. Furthermore, we incorporate optimal solutions from small instances to greatly enhance the model's reasoning capabilities. Our empirical results demonstrate that our model not only simulates but also outperforms approximation algorithms for multiple tasks, exhibiting robust generalization to larger and out-of-distribution graphs. Moreover, we highlight the framework's practical utility by integrating it with commercial solvers and applying it to real-world datasets.

## 1. Introduction

Understanding the algorithmic reasoning ability of neural networks is crucial for quantifying their expressivity and practical deployment (Łukasz Kaiser & Sutskever, 2016; Zhou et al., 2022; Sanford et al., 2024; de Luca & Fountoulakis, 2024). However, end-to-end supervised learning often struggles with generalization for algorithmic tasks. Neural Algorithmic Reasoning (NAR) (Veličković & Blundell, 2021) addresses this challenge by training neural networks to mimic operations of classical algorithms, such as

[1]Department of Computer Science, Stanford University, Stanford, CA, USA [2]Department of Management Science & Engineering, Stanford University, Stanford, CA, USA. Correspondence to: Yu He <heyu@stanford.edu>.

*Proceedings of the $42^{nd}$ International Conference on Machine Learning*, Vancouver, Canada. PMLR 267, 2025. Copyright 2025 by the author(s).

Bellman-Ford for shortest-path problems (Veličković et al., 2020). By aligning a model's architecture with an algorithm's step-wise operations, NAR enhances generalization and sample efficiency (Xu et al., 2020; 2021).

Moreover, NAR addresses a fundamental bottleneck of classical algorithms: while they guarantee correctness and interoperability, they require extensive feature engineering to compress real-world data into scalar values. By embedding algorithmic knowledge into neural models, NAR enables direct handling of structured real-world data (Deac et al., 2021; He et al., 2022; Beurer-Kellner et al., 2022; Numeroso et al., 2023). For instance, a model pre-trained with Bellman-Ford knowledge can tackle real-world transportation problems, while integrating domain-specific features such as weather conditions and traffic patterns.

Despite the success of NAR in simulating polynomial-time algorithms (Ibarz et al., 2022; Rodionov & Prokhorenkova, 2024), in particular the 30 algorithms (e.g., sorting, search, graph) from the CLRS-30 benchmark (Veličković et al., 2022) and other benchmarks (Minder et al., 2023; Markeeva et al., 2024), its extension to NP-hard problems remains an open challenge due to the difficulty in reliably generating ground-truth samplers (Veličković et al., 2022). This limitation creates a significant gap when applying NAR to real-world problems, many of which are inherently NP-hard. Addressing this gap is crucial, as the motivation behind NAR is to enable the transfer of algorithmic knowledge to tackle complex, real-world datasets effectively.

We build upon the line of NAR research that simulates classical algorithms with Graph Neural Networks (GNNs) (Xu et al., 2020; Veličković et al., 2020; Bevilacqua et al., 2023; Rodionov & Prokhorenkova, 2023; 2024; Georgiev et al., 2024). Specifically, we focus on advancing NAR into the underexplored NP-hard domain (Cappart et al., 2022; Georgiev et al., 2023c) by training GNNs to replicate approximation algorithms. Additionally, we leverage findings that NAR benefits from multi-task learning (Xhonneux et al., 2021; Ibarz et al., 2022) when trained on multiple algorithms. In particular, we harness the concept of duality, which frames problems through complementary primal and dual perspectives, a principle that has been shown to enhance NAR (Numeroso et al., 2023) in the context of Ford-Fulkerson via the max-flow min-cut theorem.

Our key contributions are as follows:

- We propose **Primal-Dual Neural Algorithmic Reasoning (PDNAR)** – a general NAR framework based on the primal-dual paradigm to learn algorithms for both polynomial-time-solvable and NP-hard tasks.
- We establish a novel alignment between primal-dual algorithms and GNNs using a bipartite representation between primal and dual variables.
- We provide theoretical proofs showing that PDNAR can exactly replicate the classical primal-dual algorithm and inherit performance guarantees.
- We go beyond algorithm replication by incorporating optimal supervision signals from small problem instances, enabling it to outperform the algorithm it is trained on[1]. To the best of our knowledge, this is the first NAR method designed to achieve this.
- We empirically validate PDNAR on synthetic algorithmic datasets, demonstrating strong generalization to larger problem instances and OOD distributions.
- We showcase PDNAR's practical utility by applying it to real-world datasets and commercial solvers.

Our code can be found at https://github.com/dransyhe/pdnar.

## 2. Related works

The most closely related work extending NAR to NP-hard domain is by Georgiev et al. (2023c), which pretrains a GNN on algorithms for polynomial-time-solvable problems (e.g., Prim's algorithm for MST) and using transfer learning to solve NP-hard problems (e.g., TSP). However, this approach lacks generality since each NP-hard problem requires carefully selecting a related polynomial-time-solvable problem and algorithm. However, our method is inherently general, directly learning approximation algorithms for NP-hard tasks and incorporating optimal solutions from small instances to enhance the model's reasoning ability.

Only one prior work has explored the role of duality in NAR. Numeroso et al. (2023) trained a GNN to replicate the Ford-Fulkerson algorithm for the max-flow (primal) and min-cut (dual) problems. By leveraging supervision signals from both problems, their model benefits from multi-task learning to achieve better performance. However, their architecture is highly specialized for Ford-Fulkerson, and both problems are polynomial-time solvable and thus benefit from strong duality. In contrast, our framework is general and applicable

---

[1]Many classical primal-dual algorithms achieve tight worst-case approximation bounds under the Unique Games Conjecture (e.g., Khot & Regev, 2008). Although worst-case limits are unlikely to be exceeded, our empirical results demonstrate improved performance in a setting beyond the worst-case.

to a range of exact and approximation algorithms.

Another line of work, Neural Combinatorial Optimization (NCO), develops neural approaches for solving NP-hard problems. However, NCO and NAR differ fundamentally. NCO focuses on learning task-specific heuristics or end-to-end optimization methods for finding (near-)optimal solutions (Dai et al., 2018; Li et al., 2018; Joshi et al., 2019; Karalias & Loukas, 2021; Wang & Li, 2023; Wenkel et al., 2024; Yau et al., 2024), whereas NAR designs neural architectures to simulate and generalize algorithmic behavior across problems, allowing them to embed algorithmic knowledge in real-world settings. While our primary focus is not NCO, our framework can serve as an algorithmically informed GNN to enhance data efficiency and generalization in supervised learning. See Appendix H for details.

## 3. Background

### 3.1. Problem statement: algorithmic reasoning

The task of algorithmic reasoning is to learn a model $\mathcal{M}$ that approximates the behavior of a target algorithm $\mathcal{A} : \mathcal{X} \to \mathcal{Y}$, where inputs $x \in \mathcal{X}$ map to outputs $y = \mathcal{A}(x) \in \mathcal{Y}$. Unlike standard function approximation, the goal is to model the sequence of intermediate states $\{S^{(t)}\}_{t=0}^{T}$ generated by $\mathcal{A}$ during execution with trajectory $\mathcal{M}^{(t)}(x) = S^{(t)}$.

In particular, we focus on simulating the primal-dual algorithm, which provides a general framework to design algorithms for both polynomial-time-solvable and NP-hard problems. We prioritize the latter due to the limited NAR research in the NP-hard domain.

### 3.2. Algorithmic tasks

We study the following algorithmic problems.

**Definition 3.1** (Minimum Vertex Cover). Let $G = (V, \mathcal{E})$ be a graph where $V$ are vertices and $\mathcal{E}$ are edges, and each vertex $v \in V$ has a non-negative weight $w_v \in \mathbb{R}^+$. A vertex cover for $G$ is a subset $C \subseteq V$ of the vertices such that for each edge $(v, u) \in \mathcal{E}$, either $v \in C$, $u \in C$, or both. The objective is to minimize the total vertex weight $\sum_{v \in C} w_v$.

**Definition 3.2** (Minimum Set Cover). Given a ground set $\mathcal{U}$ and a family of sets $\mathcal{C} \subseteq 2^{\mathcal{U}}$ with non-negative weights $w_S \in \mathbb{R}^+$ for all sets $S \in \mathcal{C}$, a set cover is a subfamily $\mathcal{C}' \subseteq \mathcal{C}$ such that $\cup_{S \in \mathcal{C}'} S = \cup_{S \in \mathcal{C}} S$. The objective is to minimize the total weight $\sum_{S \in \mathcal{C}'} w_S$.

**Definition 3.3** (Minimum Hitting Set). Given a ground set $E$ of elements $e$ with non-negative weights $w_e \in \mathbb{R}^+$ and a collection $\mathcal{T}$ of subsets $T \subseteq E$, a hitting set is a subset $A \subseteq E$ such that $A \cap T \neq \emptyset$ for every $T \in \mathcal{T}$. The objective is to minimize the total weight $\sum_{e \in A} w_e$.

$$\text{Minimize} \quad \sum_{e \in E} w_e x_e$$

$$\text{subject to} \quad \sum_{e \in T} x_e \geq 1, \quad \forall T \in \mathcal{T}$$

$$x_e \in \{0,1\}, \quad \forall e \in E$$

$$\text{Minimize} \quad \sum_{e \in E} w_e x_e$$

$$\text{subject to} \quad \sum_{e \in T} x_e \geq 1, \quad \forall T \in \mathcal{T}$$

$$x_e \geq 0, \quad \forall e \in E$$

$$\text{Maximize} \quad \sum_{T \in \mathcal{T}} y_T$$

$$\text{subject to} \quad \sum_{T: e \in T} y_T \leq w_e, \quad e \in E$$

$$y_T \geq 0, \quad \forall T \in \mathcal{T}$$

**(a) Minimum Hitting Set (MHS)**   **(b) LP relaxation of MHS**   **(c) Dual of LP relaxation of MHS**

*Figure 1.* Let $x_e \in \{0, 1\}$ for each element $e \in E$ be the variables, where $x_e = 1$ represents that element $e$ is included in the hitting set $A$, the IP formulation of MHS is shown in (a). Let $x_e \in \mathbb{R}^+$ be the primal variables, the LP relaxation of MHS is shown in (b). Let $y_T \in \mathbb{R}^+$ for each set $T \in \mathcal{T}$ be the dual variables, the dual problem of the LP relaxation of MHS is shown in (c).

---

**Algorithm 1** General primal-dual approximation algorithm

---

**Input:** $(\mathcal{T}, E, w)$: a ground set $E$ with weights $w$, a family of subsets $\mathcal{T} \subseteq 2^E$
$A \leftarrow \emptyset$; for all $e \in E, r_e \leftarrow w_e$
**while** $\exists T : A \cap T = \emptyset$ **do**
  $\mathcal{V} \leftarrow \{T : A \cap T = \emptyset\}$
  **repeat**
    **for** $T \in \mathcal{V}$ **do** $\delta_T \leftarrow \min_{e \in T} \left\{ \frac{r_e}{|\{T': e \in T'\}|} \right\}$        Uniform increase (optional):
    **for** $e \in E \setminus A$ **do** $r_e \leftarrow r_e - \sum_{T: e \in T} \delta_T$        (6.1): $\Delta \leftarrow \min_{T \in \mathcal{V}} \delta_T$
  **until** $\exists e \notin A : r_e = 0$      $\longrightarrow$    (6.2): **for** $e \in E \setminus A$ **do** $r_e \leftarrow r_e - |\{T : e \in T\}|\Delta$
  $A \leftarrow A \cup \{e : r_e = 0\}$
**end while**
**Output:** $A$

---

The minimum vertex cover problem is a foundational NP-hard problem with wide-reaching applications, where its dual problem is the well-studied maximum edge-packing problem. This primal-dual pair inspired a famous 2-approximation algorithm proposed by Hochbaum (1982) and many follow-up works. The minimum set cover problem is a generalization of vertex cover to hypergraphs, providing a more complex structural setting to evaluate algorithmic reasoning. Lastly, the hitting set is equivalent to the set cover problem, but its formulation more naturally extends to a wide range of problems, including vertex cover, Steiner tree, feedback vertex set, and many more (Goemans & Williamson, 1996).

### 3.3. A general primal-dual approximation algorithm

We now illustrate a general primal-dual approximation algorithm using the Minimum Hitting Set (MHS) problem as a concrete example. MHS can be formulated as an integer program (IP) as shown in Figure 1(a), where variables are restricted to integer values. The linear programming (LP) relaxation relaxes the integer constraints and allows variables to take continuous values, making the problem more tractable. This is illustrated in Figure 1(b).

Every LP formulation has a dual version. For MHS, Figure 1(b) is the *primal* and Figure 1(c) is the *dual*. More gener-

ally, the dual of an LP $\min_{\boldsymbol{x} \geq 0}\{\boldsymbol{c}^\top \boldsymbol{x} : \boldsymbol{A}\boldsymbol{x} \geq \boldsymbol{b}\}$ is defined as $\max_{\boldsymbol{y} \geq 0}\{\boldsymbol{y}^\top \boldsymbol{b} : \boldsymbol{A}^T \boldsymbol{y} \leq \boldsymbol{c}\}$. The *weak duality principal* states that any feasible solution to the primal problem has a larger objective value than any feasible solution to the dual problem. Based on this principle, the primal-dual framework iteratively updates both the primal and dual solutions, closing their gap and ensuring they improve in tandem.

Based on the primal-dual framework, an $\alpha$-approximation algorithm (Bar-Yehuda & Even, 1981; Hochbaum, 1982; Goemans & Williamson, 1996; Khuller et al., 1994) for the general hitting set problem was developed, where $\alpha$ is the maximal cardinality of the subsets. The pseudocode of the algorithm is shown in Algorithm 1. Given a hitting set problem $(\mathcal{T}, E, w)$, the algorithm progresses over a series of rounds. At each round, the algorithm increases some of the dual variables $y_T$ until a constraint $\sum_{T: e \in T} y_T \leq w_e$ becomes equality, at which point the element $e$ is added to the hitting set $A$. Although the algorithm does not explicitly define the dual variables $y_T$, it can be interpreted as gradually increasing the dual variables by an amount $\delta_T$ in each round, as shown in Line 1. This is implemented by defining a residual weight $r_e = w_e - \sum_{T: e \in T} y_T$, which is defined in terms of the step sizes $\delta_T$, as shown in Line 1. Once $r_e = 0$ for some $e \notin A$ (i.e. the constraint becomes tight), $e$ is added to the hitting set $A$ (Lines 1 and 1). This process is repeated until $A$ is a valid hitting set (Line 1).

**A general framework** This algorithm can be reformulated to recover many classical (exact or approximation) algorithms (Goemans & Williamson, 1996). For example, vertex cover can be seen as a hitting set problem, where element $e \in E$ corresponds to vertex $v \in V$, and subset $T \in \mathcal{T}$ corresponds to an edge that connects two vertices. This allows a direct adaptation of Algorithm 1 to solve the vertex cover problem. Moreover, Khuller et al. (1994) propose a sublinear-time vertex cover approximation algorithm which is a simple generalization of Algorithm 1 (see Appendix A). They relax the dual constraint using a parameter $\epsilon > 0$, such that a vertex $e$ is included in the cover if $r_e \leq \epsilon w_e$, instead of $r_e = 0$. This leads to a $2/(1-\epsilon)$-approximation algorithm with a runtime of $O(\ln^2 |\mathcal{T}| \ln \frac{1}{\epsilon})$. Since set cover extends vertex cover to hypergraphs, this algorithm can be adapted into an $r/(1-\epsilon)$-approximation algorithm for set cover, where $r$ is the maximal cardinality of the sets. We will later empirically show how well our framework simulates these algorithms. Moreover, the generality of the primal-dual framework is not limited by approximation algorithms for NP-hard tasks: it also recovers exact algorithms for some polynomial-time-solvable problems, such as Kruskal's minimum spanning tree algorithm (Kruskal, 1956).

**Uniform increase of dual variables** Some problems benefit from simultaneously increasing all dual variables $\delta_T$ at the same rate (Agrawal et al., 1995; Goemans & Williamson, 1995). An example is how Kruskal's algorithm greedily selects the minimum-cost edge that connects two distinct components. This corresponds to increasing the dual variables for all connected components simultaneously until there is an edge whose dual constraint becomes tight. The incorporation of this uniform increase rule is illustrated from Line 6.1 and Line 6.2 in Algorithm 1. This rule provides a more balanced approach that attends to all dual variables, allowing the framework to adapt to a broader range of algorithms.

## 4. Primal-Dual Neural Algorithmic Reasoning (PDNAR)

We now present our framework of using a GNN to simulate the general primal-dual approximation algorithm, representing the primal-dual variables as two sides in a bipartite graph (Section 4.1). We also show how the uniform increase rule can be incorporated with a virtual node that connects to all dual nodes in the bipartite graph (Section 4.2). Furthermore, we explain how we use optimal solutions from integer programming solvers as additional training signals (Section 4.4), and later show how it allows the PDNAR to surpass the performance of the approximation algorithm.

### 4.1. Architecture

We adopt the encoder-processor-decoder framework (Hamrick et al., 2018) from the neural algorithmic reasoning

blueprint (Veličković & Blundell, 2021). In this framework, the *processor* is typically a message-passing GNN (Gilmer et al., 2017), operating in a latent space to simulate a single step of algorithmic execution. The *encoder* transforms the input value (e.g., element weight) into this latent space, while the *decoder* reconstructs the final prediction from it (e.g., whether to include the element in the solution). We continue using Algorithm 1 for MHS as an example.

**Bipartite graph construction** Given a hitting set problem $(\mathcal{T}, E, w)$, we represent it as a bipartite graph with elements $e \in E$ (primal) on the left-hand side (LHS) and sets $T \in \mathcal{T}$ (dual) on the right-hand side (RHS), as illustrated in Figure 2. An edge connects an element $e$ and a set $T$ if $e \in T$. Let $\mathcal{N}(e)$ denote the set of neighbors of node $e$. As outlined in Algorithm 1, the algorithm incrementally adds elements $e$ to the hitting set $A$. When an element is added, we remove node $e$ by masking it out, along with its neighboring sets $T \in \mathcal{N}(e)$, which are now hit. Consequently, the violation set $\mathcal{V} = \{T \mid A \cap T = \emptyset\}$ consists of the remaining sets $T$ still in the graph. Once $A$ becomes a valid hitting set, the violation set $\mathcal{V}$ is empty. Next, at each timestep $t$, we let $r_e^{(t)}$ denote the residual weight of element $e$ and $d_e^{(t)}$ denote its current node degree. Therefore, the initial residual weight $r_e^{(0)}$ is defined as its cost $w_e$, and the initial degree $d_e^{(0)}$ is given by $|\{T : e \in T\}|$.

**Encoder** The architecture includes two MLP encoders, $f_r$ and $f_d$, which encode the residual node weight $r_e^{(t)}$ and node degree $d_e^{(t)}$, respectively, for each element $e \in E$. These encoders transform all features into a high-dimensional latent space for the processor:

$$\boldsymbol{h}_e^{(t)} = f_r(r_e^{(t-1)}), \qquad \boldsymbol{h}_{d_e}^{(t)} = f_d(d_e^{(t-1)}).$$

**Processor** The processor is a message-passing GNN applied to the bipartite graph. A general message-passing framework (Gilmer et al., 2017) comprises a message function $\psi_\theta$ and an update function $\phi_\theta$. The node feature $\boldsymbol{h}_v^{(t)}$ of node $v$ is transformed via

$$\boldsymbol{h}_v^{(t)} = \phi_\theta\left(\boldsymbol{h}_v^{(t)}, \bigoplus_{u \in \mathcal{N}(v)} \psi_\theta(\boldsymbol{h}_u^{(t)})\right),$$

where $\psi_\theta$ and $\phi_\theta$ are usually shallow MLPs and $\bigoplus$ is a permutation invariant function, such as sum or max. We now demonstrate how this message-passing framework is applied to the bipartite graph to simulate the primal-dual approximation algorithm.

Step (1): This step corresponds to Line 1 of Algorithm 1, where increment $\delta_T^{(t)}$ for each set $T \in \mathcal{V}$ is computed. Let $\boldsymbol{h}_T^{(t)}$ be the hidden representation of $\delta_T^{(t)}$. We aggregate

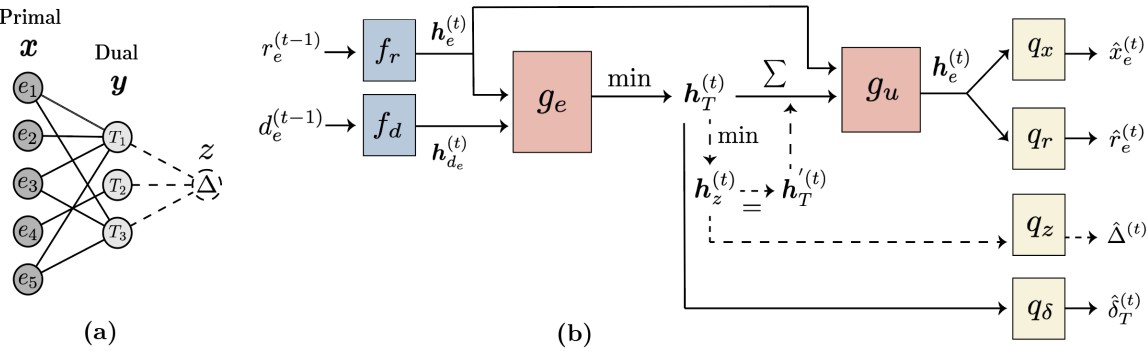

*Figure 2.* (a) Bipartite graph construction. (b) The architecture of PDNAR with the encoder, processor, and decoder colored distinctively. $\Delta$ is only used when the uniform increase rule is applied.

messages from its connected elements $e \in \mathcal{N}(T)$ using a message function $g_e$ with a min aggregation operation:

$$\boldsymbol{h}_T^{(t)} = \min_{e \in \mathcal{N}(T)} g_e(\boldsymbol{h}_e^{(t)}, \boldsymbol{h}_{d_e}^{(t)}).$$

Step (2): This step corresponds to Line 1 of Algorithm 1, where residual weight $r_e^{(t)}$ for each element $e \in E \setminus A$ is computed. Therefore, the dual variable update $\boldsymbol{h}_T^{(t)}$ is passed back to its connected elements $e$ using a sum aggregation and an update function $g_u$:

$$\boldsymbol{h}_e^{(t)} = g_u \left( \boldsymbol{h}_e^{(t)}, \sum_{T \in \mathcal{N}(e)} \boldsymbol{h}_T^{(t)} \right).$$

**Decoder** At each timestep $t$, Algorithm 1 computes three types of intermediate quantities: (1) whether to include an element $e$ in the hitting set, represented by $x_e^{(t)} \in \{0, 1\}$, (2) the residual weights of an element $r_e^{(t)}$, and (3) the increment to the dual variable $\delta_T^{(t)}$. We utilize separate MLP decoders, $q_x$, $q_r$, and $q_\delta$, to compute each of these quantities:

$$\hat{x}_e^{(t)} = q_x(\boldsymbol{h}_e^{(t)}), \quad \hat{r}_e^{(t)} = q_r(\boldsymbol{h}_e^{(t)}), \quad \hat{\delta}_T^{(t)} = q_\delta(\boldsymbol{h}_T^{(t)}).$$

**Training** Given the recurrent nature of our architecture, we apply *noisy teacher forcing* (Veličković et al., 2022) with a probability of 0.5 to determine whether to use *hints*—ground-truth values for intermediate quantities above—as inputs for the next timestep. Otherwise, the model's prediction from the previous timestep is passed on. This approach allows the model to follow its recurrent flow while reducing the risk of error propagation. The recurrent model is repeated for a maximum of $|E|$ timesteps or terminates early when the solution becomes a valid hitting set. The loss function is defined as $\mathcal{L}_{\text{algo}}^{(t)} = \mathcal{L}_{\text{BCE}}(\hat{x}_e^{(t)}, x_e^{(t)}) + \mathcal{L}_{\text{MSE}}(\hat{r}_e^{(t)}, r_e^{(t)}) + \mathcal{L}_{\text{MSE}}(\hat{\delta}_T^{(t)}, \delta_T^{(t)})$ and averaged across timesteps. During test time, if the model

output has not produced a valid hitting set, we greedily add the element $e$ with the highest $r_e/d_e$ value to the solution. Empirically, we find our model rarely requires it.

### 4.2. Uniform increase of dual variables

The uniform increase rule requires global communication among all dual variables. To achieve this, we introduce a virtual node $z$ that connects to every set $T \in \mathcal{T}$, as shown in Figure 2. Below, we describe how Step (2) in the processor is adjusted to accommodate this modification.

- Step (2.1): The virtual node aggregates all messages from the dual variables $\boldsymbol{h}_T^{(t)}$ via a min aggregation, corresponding to Line 6.1 via $\boldsymbol{h}_z^{(t)} = \min_{T \in \mathcal{T}} \boldsymbol{h}_T^{(t)}$.

- Step (2.2): The global information is passed back to dual variables with temporary $\boldsymbol{h}_T'^{(t)} = \boldsymbol{h}_z^{(t)}$, and then to the primal variables $\boldsymbol{h}_e^{(t)}$ with an update function $g_u$ and a sum aggregation. This corresponds to Line 6.2 via $\boldsymbol{h}_e^{(t)} = g_u(\boldsymbol{h}_e^{(t)}, \sum_{T \in \mathcal{N}(e)} \boldsymbol{h}_T'^{(t)})$.

The intermediate quantity $\Delta^{(t)}$ is also given by Algorithm 1. We use an additional decoder $q_\Delta$ to predict $\hat{\Delta}^{(t)} = q_\Delta(\boldsymbol{h}_z^{(t)})$ and add $\mathcal{L}_{\text{MSE}}(\hat{\Delta}^{(t)}, \Delta^{(t)})$ to the total loss $\mathcal{L}_{\text{algo}}^{(t)}$.

### 4.3. Theoretical justification

Algorithmic alignment is critical for generalization in NAR (Xu et al., 2020; 2021). We show that PDNAR can exactly replicate the behavior of Algorithm 1.

**Theorem 4.1.** *Given a hitting set problem $(\mathcal{T}, E, w)$, let $\mathcal{A}(\mathcal{T}, E, w)$ be the solution produced by Algorithm 1, which terminates after $K$ timesteps. There exists a parameter configuration $\Theta$ for a PDNAR model $\mathcal{M}_\Theta$ such that, at timestep $K$, the model output satisfies $\mathcal{M}_\Theta^{(K)}(\mathcal{T}, E, w) = \mathcal{A}(\mathcal{T}, E, w)$. Furthermore, let $(\boldsymbol{x}^{(t)}, \boldsymbol{r}^{(t)}, \boldsymbol{\delta}^{(t)}, \Delta^{(t)})$ be*

*the intermediate quantities computed by Algorithm 1 at each timestep $t$. Then, the PDNAR model satisfies $\mathcal{M}_{\Theta}^{(t)}(\mathcal{T}, E, w) = (\boldsymbol{x}^{(t)}, \boldsymbol{r}^{(t)}, \boldsymbol{\delta}^{(t)}, \Delta^{(t)})$, where $\Delta^{(t)}$ is omitted if the uniform increase rule is not applied.*

In our proof of Theorem 4.1 (Appendix B), we show that this can be achieved with a PDNAR model using 8 layers. Therefore, PDNAR inherits the approximation ratio and convergence guarantees of the primal-dual approximation algorithm it learns, as described in Corollary 4.2.

**Corollary 4.2.** *There exists a parameterization of the PDNAR model $\mathcal{M}_{\Theta}$ with 8 layers such that, after $K$ iterations, $\mathcal{M}_{\Theta}^{(K)}(\mathcal{T}, E, w)$ yields an $\alpha$-approximation to the optimal solution the MHS problem, where $\alpha = max_{T \in \mathcal{T}}(|T|)$ and $K = O(|E|)$.*

This guarantee also extends to the $2/(1 - \epsilon)$-approximation algorithm (Khuller et al., 1994) for MVC (and thus MSC).

**Corollary 4.3.** *Given $\epsilon \in (0, 1)$, there exists a parameterization of the PDNAR model $\mathcal{M}_{\Theta}$ with 8 layers such that, after $K$ iterations, $\mathcal{M}_{\Theta}^{(K)}(V, E, w)$ yields an $2/(1 - \epsilon)$-approximation to the optimal solution of the MVC problem, where $K = O(\log m \log(1/\epsilon))$ and $m = |E|$.*

### 4.4. Use of optimal solutions from solvers

We can compute optimal solutions using IP solvers for small problem instances. We use the default IP solver in `scipy` based on HiGHS (Schwendinger & Schumacher, 2023; Huangfu & Hall, 2018). These optimal solutions are used as additional training signals to guide the model toward better outcomes. However, unlike the primal-dual algorithm, which provides intermediate steps, IP solvers only produce the final optimal solution. Therefore, the corresponding loss is defined as $\mathcal{L}_{\text{optm}} = \mathcal{L}_{\text{BCE}}(\hat{x}_e^K, x_e^{\text{optm}})$, where $K$ is the final timestep. The overall loss is then the sum of the intermediate losses from the primal-dual algorithm and the optimal solution loss, given by $\mathcal{L} = \frac{1}{K} \sum_{t=1}^{K} \mathcal{L}^{(t)} \text{algo} + \mathcal{L}_{\text{optm}}$. The motivation stems from the fact that IP solvers are computationally expensive, especially for larger problem instances. By training PDNAR using optimal solutions from IP solvers on smaller problem instances—allowing it to exceed the performance of the approximation algorithm—we can leverage its generalization ability to create a cost-efficient, high-performance model for much larger problems.

## 5. Experiments

Dataset distributions and hyperparameter details of all experiments are provided in Appendices C, D, and E.

### 5.1. Synthetic algorithmic datasets

**Dataset** We evaluate PDNAR's ability to simulate the approximation algorithms for the three NP-hard algorithmic problems described in Section 3.2. The training dataset includes 1000 random graphs of size 16 for each task. We use Barabási-Albert graphs for vertex cover. For set cover and hitting set, we generate Barabási-Albert bipartite graphs with a preferential attachment parameter $b = 5$. Each test set consists of 100 graphs and is repeated for 10 seeds.

**Baselines** We compare with three types of baselines. (i) GNNs for end-to-end node classification trained with optimal labels: GIN (Xu et al., 2019) and GAT (Veličković et al., 2018). (ii) NAR models to simulate the approximation algorithm: NAR (MPNN) (Veličković et al., 2020) and the more expressive TripletMPNN (Ibarz et al., 2022) at the cost of training efficiency. Both models do not use the bipartite representation, and therefore are trained only on intermediate states of primal variables. (iii) Variations of PDNAR: No algo (trained without intermediate supervision from the algorithm) and no optm (trained without optimal solutions). Additionally, PDNAR's aggregation strategy is specifically tailored to align with the algorithm's structure. We test alternative mean and max aggregation methods.

Table 1 shows PDNAR achieves the best performance. By combining losses from intermediate steps of the approximation algorithm and optimal solutions from IP solvers, PDNAR yields the lowest ratios across all test cases. Moreover, PDNAR's performance remains stable across different graph sizes, indicating strong generalization to larger problems. Comparisons with other baselines (such as "No optm") show that incorporating supervision from optimal solutions improves the quality of final predictions, allowing PDNAR to outperform the primal-dual algorithm it was designed to simulate. In contrast, training without intermediate steps (such as "No algo") leads to a significant drop in generalization, highlighting that the primal-dual algorithm provides critical reasoning capabilities beyond merely learning optimal solution patterns.

Our model is a general NAR framework for a wide range of algorithms that uses the primal-dual paradigm. In Table 1, we see PDNAR is effective across all three tasks. For vertex cover, we demonstrate its applicability to graph-based algorithms. In set cover, we extend the framework to hypergraphs using a bipartite structure, highlighting its flexibility beyond traditional graph settings. Additionally, the uniform increase rule proves effective for hitting set, which can be instantiated to a range of exact and approximation algorithms. Notably, these results suggest that our approach may achieve even better performance when trained on more challenging instances where the optimal solutions significantly outperform those of approximation algorithms.

*Table 1.* Model-to-algorithm weight ratio trained on 16-node graphs and tested on larger graphs, calculated by the sum of weights from the model solution divided by the sum of weights from the algorithm solution ($w_{\mathrm{model}}/w_{\mathrm{algo}}$). Smaller is better for minimization tasks.

|  | Model | 16 (1x) | 32 (2x) | 64 (4x) | 128 (8x) | 256 (16x) | 512 (32x) | 1024 (64x) |
|---|---|---|---|---|---|---|---|---|
| MVC | GIN | $0.987 \pm 0.011$ | $1.040 \pm 0.041$ | $1.097 \pm 0.059$ | $1.087 \pm 0.061$ | $1.109 \pm 0.073$ | $1.120 \pm 0.083$ | $1.116 \pm 0.083$ |
|  | GAT | $0.962 \pm 0.106$ | $1.039 \pm 0.085$ | $1.072 \pm 0.099$ | $1.071 \pm 0.085$ | $1.108 \pm 0.106$ | $1.114 \pm 0.096$ | $1.125 \pm 0.082$ |
|  | NAR | $0.998 \pm 0.013$ | $0.999 \pm 0.012$ | $1.005 \pm 0.011$ | $1.002 \pm 0.012$ | $1.009 \pm 0.015$ | $1.013 \pm 0.012$ | $1.018 \pm 0.013$ |
|  | TripletMPNN | $0.982 \pm 0.015$ | $0.986 \pm 0.015$ | $0.991 \pm 0.014$ | $0.995 \pm 0.020$ | $1.000 \pm 0.013$ | $1.001 \pm 0.019$ | $1.005 \pm 0.019$ |
|  | No algo | $1.142 \pm 0.038$ | $1.115 \pm 0.027$ | $1.110 \pm 0.038$ | $1.099 \pm 0.032$ | $1.091 \pm 0.034$ | $1.099 \pm 0.036$ | $1.095 \pm 0.038$ |
|  | No optm | $0.995 \pm 0.004$ | $1.001 \pm 0.004$ | $1.001 \pm 0.005$ | $0.998 \pm 0.009$ | $0.998 \pm 0.009$ | $0.998 \pm 0.011$ | $0.994 \pm 0.011$ |
|  | PDNAR (mean) | $1.031 \pm 0.022$ | $1.062 \pm 0.031$ | $1.079 \pm 0.039$ | $1.107 \pm 0.047$ | $1.107 \pm 0.046$ | $1.122 \pm 0.049$ | $1.126 \pm 0.055$ |
|  | PDNAR (max) | $0.968 \pm 0.014$ | $1.011 \pm 0.012$ | $1.003 \pm 0.012$ | $1.005 \pm 0.014$ | $1.006 \pm 0.013$ | $1.010 \pm 0.015$ | $1.007 \pm 0.013$ |
|  | PDNAR | $\mathbf{0.943} \pm 0.004$ | $\mathbf{0.957} \pm 0.002$ | $\mathbf{0.966} \pm 0.002$ | $\mathbf{0.958} \pm 0.002$ | $\mathbf{0.958} \pm 0.002$ | $\mathbf{0.958} \pm 0.002$ | $\mathbf{0.957} \pm 0.002$ |
| MSC | No algo | $1.028 \pm 0.016$ | $1.025 \pm 0.014$ | $1.010 \pm 0.017$ | $1.017 \pm 0.020$ | $1.012 \pm 0.018$ | $1.008 \pm 0.023$ | $1.006 \pm 0.027$ |
|  | No optm | $1.008 \pm 0.006$ | $1.008 \pm 0.008$ | $0.997 \pm 0.008$ | $0.992 \pm 0.006$ | $0.981 \pm 0.004$ | $0.973 \pm 0.002$ | $0.975 \pm 0.002$ |
|  | PDNAR | $\mathbf{0.979} \pm 0.003$ | $\mathbf{0.918} \pm 0.013$ | $\mathbf{0.947} \pm 0.009$ | $\mathbf{0.915} \pm 0.005$ | $\mathbf{0.920} \pm 0.007$ | $\mathbf{0.915} \pm 0.006$ | $\mathbf{0.913} \pm 0.003$ |
| MHS | No algo | $1.047 \pm 0.008$ | $1.050 \pm 0.006$ | $1.049 \pm 0.007$ | $1.036 \pm 0.016$ | $1.065 \pm 0.023$ | $1.122 \pm 0.031$ | $1.256 \pm 0.036$ |
|  | No optm | $1.002 \pm 0.000$ | $1.008 \pm 0.003$ | $0.994 \pm 0.006$ | $0.999 \pm 0.007$ | $1.005 \pm 0.010$ | $1.015 \pm 0.013$ | $1.053 \pm 0.018$ |
|  | PDNAR | $\mathbf{0.989} \pm 0.002$ | $\mathbf{0.982} \pm 0.005$ | $\mathbf{0.985} \pm 0.005$ | $\mathbf{0.965} \pm 0.006$ | $\mathbf{0.990} \pm 0.008$ | $\mathbf{0.996} \pm 0.009$ | $\mathbf{1.027} \pm 0.020$ |

*Table 2.* Model-to-algorithm weight ratio trained on B-A (bipartite) graphs and tested on OOD graph families.

|  |  | 16 (1x) | 32 (2x) | 64 (4x) | 128 (8x) | 256 (16x) | 512 (32x) | 1024 (64x) |
|---|---|---|---|---|---|---|---|---|
| MVC | E-R | $0.955 \pm 0.004$ | $0.934 \pm 0.004$ | $0.934 \pm 0.005$ | $0.950 \pm 0.004$ | $0.992 \pm 0.007$ | $0.989 \pm 0.008$ | $0.993 \pm 0.008$ |
|  | Star | $0.966 \pm 0.004$ | $0.979 \pm 0.003$ | $0.977 \pm 0.005$ | $0.982 \pm 0.006$ | $0.989 \pm 0.005$ | $0.992 \pm 0.007$ | $0.998 \pm 0.006$ |
|  | Lobster | $0.971 \pm 0.003$ | $0.965 \pm 0.005$ | $0.972 \pm 0.004$ | $0.960 \pm 0.005$ | $0.970 \pm 0.008$ | $0.966 \pm 0.009$ | $0.966 \pm 0.009$ |
|  | 3-Con | $0.974 \pm 0.002$ | $0.965 \pm 0.002$ | $0.965 \pm 0.005$ | $0.957 \pm 0.006$ | $0.963 \pm 0.006$ | $0.962 \pm 0.008$ | $0.961 \pm 0.009$ |
| MSC | $b = 3$ | $0.943 \pm 0.008$ | $0.941 \pm 0.005$ | $0.916 \pm 0.008$ | $0.918 \pm 0.006$ | $0.924 \pm 0.003$ | $0.929 \pm 0.003$ | $0.922 \pm 0.004$ |
|  | $b = 8$ | $0.969 \pm 0.010$ | $0.950 \pm 0.015$ | $0.955 \pm 0.016$ | $0.940 \pm 0.013$ | $0.944 \pm 0.010$ | $0.941 \pm 0.017$ | $0.943 \pm 0.012$ |
| MHS | $b = 3$ | $0.988 \pm 0.002$ | $0.995 \pm 0.003$ | $0.985 \pm 0.004$ | $0.982 \pm 0.005$ | $0.982 \pm 0.002$ | $1.008 \pm 0.015$ | $1.005 \pm 0.036$ |
|  | $b = 8$ | $0.979 \pm 0.002$ | $0.978 \pm 0.005$ | $0.973 \pm 0.008$ | $0.960 \pm 0.003$ | $0.983 \pm 0.010$ | $1.008 \pm 0.015$ | $1.014 \pm 0.018$ |

## 5.2. Size and OOD generalization

**Dataset** We evaluate the model's generalization performance on larger and OOD graph families, using the same training set as before. For the vertex cover problem, we generate three OOD test sets comprising Erdős–Rényi (E-R), Star, Lobster, and 3-connected planar (3-Con) graphs. These graph types pose unique challenges for the vertex cover problem due to their distinct structural properties. For set cover and hitting set, we vary the preferential attachment parameter $b$ to 3 and 8 to generate OOD bipartite graphs.

Table 1 demonstrates that PDNAR, trained on 16-node graphs, scales robustly to larger graphs of up to 1024 nodes for MVC. We highlight that prior NAR research typically evaluates up to 128 nodes. Furthermore, Table 2 shows PDNAR's ability to generalize to graphs from OOD families. Notably, these graph types can exhibit significantly different optimal sizes: Erdős–Rényi graphs require an average of 80% of nodes, while Star graphs need only 15% (see Table 5), highlighting the strong generalization ability of PDNAR. Obtaining optimal solutions for small instances is fast and efficient, even for hard problems, particularly with integer programming solvers like Gurobi (Gurobi Optimization, LLC, 2024). However, the computational complexity increases exponentially as the problem size grows. This underscores the key strength of PDNAR — we can train it efficiently using small problem instances and apply it to much larger, unseen problems that are computationally expensive to solve.

## 5.3. Real-world datasets

We now present a practical use case of our model. One of the key strengths of NAR is its ability to embed algorithmic knowledge in neural networks to tackle real-world challenges. Previous works applied NAR to reinforcement learning (Deac et al., 2021), brain vessels (Numeroso et al., 2023), and computer network configuration (Beurer-Kellner et al., 2022). Traditional algorithms designed for specific problems face a significant limitation: they cannot be directly applied to real-world data without substantial preprocessing. Indeed, real-world graphs generally consist of high-dimensional features rather than simple scalar weights—the standard input for most traditional algorithms. As a result, applying traditional algorithms would necessitate extensive feature engineering to derive scalar weights, which may lead to the loss of crucial information embedded in the high-dimensional features.

In contrast, PDNAR overcomes this limitation by integrating a feature encoder that learns to estimate vertex weights

directly from raw data, eliminating the need for manual pre-processing. We showcase such an application using the Airports datasets (Brazil, Europe, USA) (Ribeiro et al., 2017). In these graphs, the nodes represent airports, and the edges represent commercial flight routes. The goal is to predict the activity level of each airport, where vertex cover solutions can be highly valuable in predicting node influence.

**Architecture and baselines**   We follow similar evaluation settings as Numeroso et al. (2023). We use three base models to perform node classification: GCN (Kipf & Welling, 2017), GAT (Veličković et al., 2018), and GraphSAGE (Hamilton et al., 2017). Then, we use PDNAR to produce embeddings to be concatenated with the base model's outputs before a final classification layer. We use a pretrained PDNAR on B-A graphs of size 16, keeping only the processor and degree encoder. We train a new encoder that learns to map new node features into the shared latent space of the processor, enabling it to replicate the vertex cover problem-solving behavior on airport data. The pretrained components are frozen, and a single message-passing step is performed. We compare PDNAR's embeddings against several baselines: Node2Vec (Grover & Leskovec, 2016) and a degree encoder. The latter helps to show that PDNAR captures more complex information beyond node degrees. We also compare with two positional encodings: LapPE (Dwivedi et al., 2022a) and RWPE (Dwivedi et al., 2022b).

Table 3 shows that PDNAR achieves significant improvements in all datasets. Note that Node2Vec has an additional advantage by directly training on the graphs to produce the embeddings. PDNAR's superior performance over the degree encoder also indicates that it captures more complex information by integrating both learned node weights from features and degree information to replicate vertex cover. This showcases the important practical value of NAR in embedding algorithmic knowledge in real-world applications, overcoming the bottleneck of traditional algorithms.

## 5.4. Commercial optimization solvers

Another practical use case for PDNAR is to warm start large-scale commercial solvers, such as Gurobi (Gurobi Optimization, LLC, 2024), by initializing variables with its predictions. The motivation is that providing a starting point closer to the optimal solution can lead to faster solving times and improved efficiency.

**Dataset**   We evaluate the vertex cover problem by comparing Gurobi's default initialization to warm starts using solutions from the primal-dual algorithm and our model. Trained on 1000 B-A graphs of size 16, the model generates solutions for random larger B-A graphs, with 100 graphs per size. Gurobi's default parameters are used, with a thread count of 1 and a 1-hour time limit.

*Table 3.* Accuracy (%) for the three Airports datasets comparing embeddings generated using different methods.

|  | Brazil | Europe | USA |
|---|---|---|---|
| GCN | $58.89_{\pm 17.72}$ | $63.87_{\pm 10.74}$ | $77.48_{\pm 2.28}$ |
| GCN+LapPE | $61.48_{\pm 17.23}$ | $73.50_{\pm 3.83}$ | $78.11_{\pm 2.94}$ |
| GCN+RWPE | $58.15_{\pm 20.15}$ | $73.25_{\pm 4.47}$ | $77.56_{\pm 3.28}$ |
| GCN+Degree | $71.48_{\pm 15.53}$ | $70.37_{\pm 4.16}$ | $79.25_{\pm 2.16}$ |
| GCN+N2V | $73.33_{\pm 5.44}$ | $74.75_{\pm 5.12}$ | $79.54_{\pm 2.19}$ |
| GCN+PDNAR | $\mathbf{81.11}_{\pm 9.46}$ | $\mathbf{76.88}_{\pm 5.10}$ | $\mathbf{82.82}_{\pm 3.06}$ |
| GAT | $60.13_{\pm 15.68}$ | $65.85_{\pm 9.68}$ | $80.59_{\pm 2.43}$ |
| GAT+LapPE | $62.22_{\pm 19.23}$ | $71.00_{\pm 4.57}$ | $79.54_{\pm 2.57}$ |
| GAT+RWPE | $66.30_{\pm 21.24}$ | $73.87_{\pm 3.93}$ | $82.39_{\pm 1.99}$ |
| GAT+Degree | $66.30_{\pm 12.27}$ | $77.75_{\pm 4.10}$ | $82.82_{\pm 2.08}$ |
| GAT+N2V | $75.56_{\pm 9.82}$ | $75.87_{\pm 5.03}$ | $82.65_{\pm 1.58}$ |
| GAT+PDNAR | $\mathbf{84.44}_{\pm 8.89}$ | $\mathbf{81.25}_{\pm 4.51}$ | $\mathbf{85.13}_{\pm 1.79}$ |
| SAGE | $44.82_{\pm 17.40}$ | $61.82_{\pm 10.38}$ | $78.07_{\pm 3.39}$ |
| SAGE+LapPE | $62.59_{\pm 22.98}$ | $64.12_{\pm 21.72}$ | $79.50_{\pm 2.02}$ |
| SAGE+RWPE | $50.74_{\pm 25.23}$ | $73.38_{\pm 3.54}$ | $80.80_{\pm 2.65}$ |
| SAGE+Degree | $74.44_{\pm 12.74}$ | $75.50_{\pm 9.19}$ | $81.53_{\pm 2.18}$ |
| SAGE+N2V | $80.00_{\pm 6.46}$ | $76.12_{\pm 3.75}$ | $83.15_{\pm 3.20}$ |
| SAGE+PDNAR | $\mathbf{85.56}_{\pm 7.49}$ | $\mathbf{80.75}_{\pm 2.38}$ | $\mathbf{83.28}_{\pm 2.04}$ |

**Metrics**   We report the average solving times of all cases when optimal solutions are found (Solve time). We also measure the average computation time to generate the warm-start solutions using the primal-dual algorithm and the model per graph (Compute time). Results are recorded in seconds and averaged across 5 seeds.

*Table 4.* Performance of warm starting Gurobi using solutions from a PDNAR trained on 16-node graphs and solutions from the primal-dual approximation algorithm.

| #Nodes | Method | Solve time (optimal found) | Compute time |
|---|---|---|---|
| 500 | None | $78.97_{\pm 2.24}$ | - |
|  | Algorithm | $76.69_{\pm 1.54}$ | 0.20 |
|  | PDNAR | $\mathbf{72.61}_{\pm 2.13}$ | 0.02 |
| 600 | None | $230.84_{\pm 19.59}$ | - |
|  | Algorithm | $227.83_{\pm 19.32}$ | 0.30 |
|  | PDNAR | $\mathbf{209.52}_{\pm 14.15}$ | 0.03 |
| 750 | None | $299.01_{\pm 23.53}$ | - |
|  | Algorithm | $300.12_{\pm 22.89}$ | 0.39 |
|  | PDNAR | $\mathbf{292.92}_{\pm 18.61}$ | 0.03 |

Table 4 shows that PDNAR outperforms both the default initialization and the approximation algorithm in all cases, achieving the fastest mean solving time. The improvement from using the model over the algorithm is greater than that of the algorithm over no warm start. Additionally, the model's inference time is nearly 10 times faster than the approximation algorithm's computation time. This demonstrates a practical use case: by simulating an approximation algorithm and leveraging optimal solutions on small

instances, the model generates high-quality solutions for larger problems, improving efficiency for large-scale commercial solvers, such as Gurobi.

## 6. Conclusions

We propose a general NAR framework using the primal-dual paradigm. Our approach can simulate approximation algorithms for NP-hard problems, greatly extending NAR's capability beyond the polynomial-time-solvable domain. Furthermore, we leverage both the intermediate states generated by the primal-dual algorithm and optimal solutions obtained from integer programming solvers on small problem instances, which can be obtained efficiently. While intermediate algorithmic steps provides a foundation for reasoning, incorporating optimal solutions enables the model to surpass the algorithm's performance. Empirical results demonstrate that our framework is effective and robust, showing strong generalization to larger problems and OOD distributions. Additionally, we present two practical applications: generating algorithmically informed embeddings for real-world datasets and warm-starting commercial solvers.

**Limitation and future work**   The primal-dual framework underlies many algorithmic problems, making our approach broadly applicable. Our architecture is currently designed for the hitting set problem, which can be reformulated as many other algorithmic problems. However, problems that do not reduce to hitting set may require architectural extensions. For instance, the uncapacitated facility location problem involves two types of dual variables that need specialized handling. Other advanced techniques (Williamson & Shmoys, 2011), such as selectively updating dual variables to improve scalability, can further extend our framework to a broader class of approximation algorithms. Future work could also explore the multi-task learning setting (Ibarz et al., 2022) in PDNAR by training on multiple approximation algorithms simultaneously.

## Impact Statement

This paper presents work whose goal is to advance the field of Machine Learning. There are many potential societal consequences of our work, none of which we feel must be specifically highlighted here.

## Acknowledgement

We thank the anonymous reviewers for their valuable feedback on this manuscript. This work was supported in part by NSF grant CCF-2338226.

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

# A. Additional details of vertex cover and set cover

## A.1. Primal-dual pair: vertex cover and edge packing

Given a graph $G = (V, E)$, where $V$ are vertices and $E$ are edges, each vertex $v \in V$ has a non-negative weight $w : V \to \mathbb{R}^+$.

**Definition A.1** (Minimum vertex cover). A vertex-cover for $G$ is a subset $C \subseteq V$ of the vertices such that for each edge $(v, u) \in E$, either $v \in C$, $u \in C$, or both. The objective is to minimize the total vertex weight $\sum_{v \in C} w(v)$.

**Definition A.2** (Maximum edge packing). An edge-packing is an assignment $p : E \to \mathbb{R}^+$ of non-negative weights to the edges $e \in E$, such that for any vertex $v \in V$, the total weight $\sum_{e:v \in e} p(e)$ assigned to the edges $e$ that are incident to $v$ is at most $w(v)$. The objective is to maximize the total edge weight $\sum_{e \in E} p(e)$.

The edge packing problem is the dual of the LP relaxation of the vertex cover problem, which also has many practical implications, such as resource allocation. Because of this relationship, the primal-dual pair becomes key problems for studying approximation algorithms and the primal-dual framework. Let $x_v \in \{0, 1\}$ indicate whether each vertex $v \in V$ is in the cover, and $y_e \in \mathbb{R}^+$ be the non-negative weight assigned to each edge $e \in E$. The two problems can be formulated as:

$$
\begin{aligned}
&\text{Min} \quad \sum_{v \in V} w(v) x_v \\
&\text{sub. to} \quad x_u + x_v \geq 1, \ \forall e = (u, v) \in E \\
&\qquad\qquad x_v \in \{0, 1\}, \ \forall v \in V
\end{aligned}
\qquad
\begin{aligned}
&\text{Max} \quad \sum_{e \in E} y_e \\
&\text{sub. to} \quad \sum_{e:v \in e} y_e \leq w(v), \ \forall v \in V \\
&\qquad\qquad y_e \geq 0, \ \forall e \in E.
\end{aligned}
$$

## A.2. Pseudocode of MVC algorithm

A $2/(1-\epsilon)$-approximation algorithm for the minimum vertex cover (MVC) problem was proposed by Khuller et al. (1994). It can be interpreted as an instantiation of the general primal-dual approximation algorithm without uniform increase (Algorithm 1). In the following, we give the original algorithm as illustrated in the original paper (Khuller et al., 1994).

Intuitively, the algorithm maintains a packing $p$ and partial cover $C_p = \{v \in V : p(E(v)) \geq (1 - \epsilon) w(v)\}$, and gradually increases the edge weights $p(e)$ as much as possible. When the constraint on the residual vertex weight is met, a vertex $v$ is removed and added to the cover $C_p$. The process iterates until $p$ is $\epsilon$-maximal and $C_p$ is a cover. Let $E_p(v)$ be the set of remaining edges incident to vertex $v$, $d_p(v) = |E_p(v)|$ be the degree, and $w_p(v) = w(v) - p(E(v))$ be the residual weight. The following is a pseudocode of the algorithm as described in Khuller et al. (1994).

---

**Algorithm 2** COVER($G = (V, E), w, \epsilon$)

---

1: **for** $v \in V$ **do**
2: $\quad w_p(v) \leftarrow w(v); \quad E_p(v) \leftarrow E(v); \quad d_p(v) \leftarrow |E(v)|$
3: **end for**
4: **while** edges remain **do**
5: $\quad$ **for** each remaining edge $(u, v)$ **do**
6: $\quad\quad \delta((u, v)) \leftarrow \min\left(\frac{w_p(u)}{d_p(u)}, \frac{w_p(v)}{d_p(v)}\right)$
7: $\quad$ **end for**
8: $\quad$ **for** each remaining vertex $v$ **do**
9: $\quad\quad w_p(v) \leftarrow w_p(v) - \sum_{e \in E_p(v)} \delta(e)$
10: $\quad\quad$ **if** $w_p(v) \leq \epsilon \cdot w(v)$ **then**
11: $\quad\quad\quad$ delete $v$ and its incident edges; update $E_p(\cdot)$ and $d_p(\cdot)$
12: $\quad\quad$ **end if**
13: $\quad$ **end for**
14: **end while**
15: **Output:** the set of deleted vertices

---

### A.3. Primal-dual pair: Set cover and element packing

The minimum set cover (MSC) problem is a generalization of MVC to hypergraphs. Similar to the edge packing, the element packing is the dual of the LP relaxation of the set cover problem.

**Definition A.3** (Minimum Set Cover). Given a universe $\mathcal{U}$ and a family of sets $\mathcal{C} \subseteq 2^{\mathcal{U}}$ with non-negative weights $w : \mathcal{C} \to \mathbb{R}^+$, a *set cover* is a subfamily $\mathcal{C}' \subseteq \mathcal{C}$ such that $\cup_{S \in \mathcal{C}'} S = \cup_{S \in \mathcal{C}} S$. The objective is to minimize the total weight $\sum_{S \in \mathcal{C}'} w(S)$.

**Definition A.4** (Maximum Element Packing). An element-packing is an assignment $p : \mathcal{U} \to \mathbb{R}^+$ of non-negative weights to the elements $e \in \mathcal{U}$, such that for any set $S \in C$, the total weight $\sum_{e \in S} p(e)$ is at most $w(S)$. The objective is to maximize the total element weight $\sum_{e \in \mathcal{U}} p(e)$.

Let $x_S \in \{0, 1\}$ indicate whether each set $S \in \mathcal{C}$ is in the cover $\mathcal{C}'$, and $y_e \in \mathbb{R}^+$ be the non-negative weight assigned to each element $e \in \mathcal{U}$. The two problems can be formulated as:

$$
\begin{aligned}
&\text{Min} &&\sum_{S \in \mathcal{C}} w(S) x_S & &\text{Max} &&\sum_{e \in \mathcal{U}} y_e \\
&\text{sub. to} &&\sum_{S : e \in S} x_S \geq 1, \ \forall e \in \mathcal{U} & &\text{sub. to} &&\sum_{e \in S} y_e \leq w(S), \ \forall S \in \mathcal{C} \\
& &&x_S \in \{0, 1\}, \ \forall S \in \mathcal{C} & & &&y_e \geq 0, \ \forall e \in \mathcal{U}.
\end{aligned}
$$

### A.4. Pseudocode of MSC Algorithm

The above algorithm can be extended for set cover as an $r/(1 - \epsilon)$-approximation algorithm, where $r$ is the maximal cardinality of sets. The following pseudocode is the adapted approximation algorithm to solve vertex cover on a hypergraph.

---

**Algorithm 3** COVER$(G = (V, E), w, \epsilon)$

---

1: **for** $v \in V$ **do**
2: $\quad w_p(v) \leftarrow w(v); \quad E_p(v) \leftarrow E(v); \quad d_p(v) \leftarrow |E(v)|$
3: **end for**
4: **while** edges remain **do**
5: $\quad$ **for** each remaining edge $e$ **do**
6: $\quad\quad \delta(e) \leftarrow \min_{v \in e} \left( \frac{w_p(v)}{d_p(v)} \right)$
7: $\quad$ **end for**
8: $\quad$ **for** each remaining vertex $v$ **do**
9: $\quad\quad w_p(v) \leftarrow w_p(v) - \sum_{e \in E_p(v)} \delta(e)$
10: $\quad\quad$ **if** $w_p(v) \leq \epsilon \cdot w(v)$ **then**
11: $\quad\quad\quad$ delete $v$ and its incident edges; update $E_p(\cdot)$ and $d_p(\cdot)$
12: $\quad\quad$ **end if**
13: $\quad$ **end for**
14: **end while**
15: **Output:** the set of deleted vertices

---

## B. Proof of Theorem 4.1

Given a hitting set problem $(\mathcal{T}, E, w)$, we construct a bipartite graph $B$ as explained in Section 4.1. Let $B^{(t)}$ denote the bipartite graph at timestep $t$, then $B^{(0)} = B$. If we remove a node $e$ and its incident edges from the graph when an element $e$ is included in the hitting set at timestep $t$, an action denoted by $x_e^{(t)} = 1$, then the bipartite graph $B^{(t)}$ changes accordingly. Therefore, the degrees of primal nodes $\boldsymbol{d}^{(t)}$ can be computed via $\boldsymbol{d}^{(t)} = f(\mathcal{T}, E, \max_{t' \in [0,...,t]}(\boldsymbol{x}^{(t')}))$ for some function $f$, where the max function is applied element-wise. WLOG, let $n$ denote the hidden dimension. We can now prove the theorem using mathematical induction.

1. Base case ($t = 0$): This is true because the inputs $\mathcal{M}_{\Theta}^{(0)}(\mathcal{T}, E, w) = (\boldsymbol{0}, \{w_e : e \in E\}, \boldsymbol{0}, 0) = (\boldsymbol{x}^{(0)}, \boldsymbol{r}^{(0)}, \boldsymbol{\delta}^{(0)}, \Delta^{(0)})$.

2. Induction step ($t > 0$): To formulate the strong induction hypothesis, let $(\boldsymbol{x}^{(t')}, \boldsymbol{r}^{(t')}, \boldsymbol{\delta}^{(t')}, \Delta^{(t')})$ be the intermediate quantites computed by Algorithm 1 for each timestep $t' \in [0, ..., t-1]$, assume $\mathcal{M}_{\Theta}^{(t')}(\mathcal{T}, E, w) = (\boldsymbol{x}^{(t')}, \boldsymbol{r}^{(t')}, \boldsymbol{\delta}^{(t')}, \Delta^{(t')})$. We now prove that $\mathcal{M}_{\Theta}^{(t)}(\mathcal{T}, E, w) = (\boldsymbol{x}^{(t)}, \boldsymbol{r}^{(t)}, \boldsymbol{\delta}^{(t)}, \Delta^{(t)})$.

The inputs for the $t^{\text{th}}$ step of our recurrent model are the outputs from the $(t-1)^{\text{th}}$ step. By the induction hypothesis, the model output $\hat{\boldsymbol{r}}^{(t-1)} = \boldsymbol{r}^{(t-1)}$. Furthermore, since $\boldsymbol{d}^{(t)} = f(\mathcal{T}, E, \max_{t' \in [0, ..., t]}(\boldsymbol{x}^{(t)'}))$ for some function $f$, and the model outputs satisfy $\hat{\boldsymbol{x}}^{(t)'} = \boldsymbol{x}^{(t)'}$ for all $t' \in [0, ..., t-1]$ via the strong induction hypothesis, we have $\hat{\boldsymbol{d}}^{(t-1)} = \boldsymbol{d}^{(t-1)}$. We take the natural logarithmic form of $\boldsymbol{r}^{(t-1)}$ and $\boldsymbol{d}^{(t-1)}$ as inputs to the encoders.

Both encoders $f_r$ and $f_d$ are MLPs. We define the weight of $f_r$ as $W_{f_r} = [1, 0, ..., 0] \in \mathbb{R}^{n \times 1}$. We define the weight of $f_d$ as $W_{f_d} = [1, 0, ..., 0] \in \mathbb{R}^{n \times 1}$. Therefore,

$$\begin{aligned}
\boldsymbol{h}_e^{(t)} &= f_r(r_e^{(t-1)}) = W_{f_r} \ln r_e^{(t-1)} \\
&= [\ln r_e^{(t-1)}, 0, ..., 0] \in \mathbb{R}^{n \times 1} \\
\boldsymbol{h}_{d_e}^{(t)} &= f_d(d_e^{(t-1)}) \\
&= W_{f_d} \ln d_e^{(t-1)} \\
&= [\ln d_e^{(t-1)}, 0, ..., 0] \in \mathbb{R}^{n \times 1}
\end{aligned}$$

The first step performs message-passing from the primal node representations $\boldsymbol{h}_e^{(t)}$ to the dual nodes $\boldsymbol{h}_T^{(t)}$ via $\boldsymbol{h}_T^{(t)} = \min_{e \in \mathcal{N}(T)} g_e(\boldsymbol{h}_e^{(t)}, \boldsymbol{h}_{d_e}^{(t)})$. Let $g_e$ be an MLP with ELU activation, then we set $W_{g_e} = [1, 0, ..., 0, -1, 0, ..., 0]^\top \in \mathbb{R}^{1 \times 2n}$, $b_{g_e} = [1] \in \mathbb{R}^1$, and $W'_{g_e} = [1, 0, ..., 0] \in \mathbb{R}^{n \times 1}$ thus:

$$\begin{aligned}
g_e(\boldsymbol{h}_e'^{(t)}) &= W'_{g_e} \left( \text{ELU} \left( W_{g_e}[\boldsymbol{h}_e^{(t)} \| \boldsymbol{h}_{d_e}^{(t)}] \right) + b_{g_e} \right) \\
&= W'_{g_e} \Big( \text{ELU}([1, 0, ..., 0, -1, 0, ..., 0]^\top \\
&\quad [\ln r_e^{(t-1)}, 0, ..., 0, \ln d_e^{(t-1)}, 0, ..., 0]) + 1 \Big) \\
&= W'_{g_e} \left( \text{ELU} \left( \ln r_e^{(t-1)} - \ln d_e^{(t-1)} \right) + 1 \right) \\
&= W'_{g_e} \left( \text{ELU} \left( \ln \frac{r_e^{(t-1)}}{d_e^{(t-1)}} \right) + 1 \right)
\end{aligned}$$

Since $\text{ELU}(x) = e^x - 1$ if $x \leq 0$, and $\ln \frac{r_e^{(t-1)}}{d_e^{(t-1)}} \leq 0$,

$$\begin{aligned}
g_e(\boldsymbol{h}_e'^{(t)}) &= W'_{g_e} \left( \exp \left( \ln \frac{r_e^{(t-1)}}{d_e^{(t-1)}} \right) - 1 + 1 \right) \\
&= [1, 0, ..., 0] \frac{r_e^{(t-1)}}{d_e^{(t-1)}} \\
&= [\frac{r_e^{(t-1)}}{d_e^{(t-1)}}, 0, ..., 0] \in \mathbb{R}^{n \times 1}
\end{aligned}$$

Therefore, the hidden representation $\boldsymbol{h}_T^{(t)}$ for each set $T \in \mathcal{T}$, is:

$$\begin{aligned}
\boldsymbol{h}_T^{(t)} &= \min_{e \in \mathcal{N}(T)} g_e(\boldsymbol{h}_e'^{(t)}) \\
&= \min_{e \in \mathcal{N}(T)} [\frac{r_e^{(t-1)}}{d_e^{(t-1)}}, 0, ..., 0]
\end{aligned}$$

---

**Algorithm 4** General primal-dual approximation algorithm

---

1: **Input:** $(\mathcal{T}, E, w)$: a ground set $E$ with weights $w$, and a family of subsets $\mathcal{T} \subseteq 2^E$
2: $A \leftarrow \emptyset$;  **for all** $e \in E$: $r_e \leftarrow w_e$
3: **while** $\exists T \in \mathcal{T} : A \cap T = \emptyset$ **do**
4:   $\mathcal{V} \leftarrow \{T \in \mathcal{T} : A \cap T = \emptyset\}$
5:   **repeat**
6:     **for each** $T \in \mathcal{V}$ **do** $\delta_T \leftarrow \min_{e \in T} \left\{ \frac{r_e}{|\{T' \in \mathcal{T}: e \in T'\}|} \right\}$    Uniform increase (optional):
7:       **for each** $e \in E \setminus A$ **do** $r_e \leftarrow r_e - \sum_{T:e \in T} \delta_T$  $\longrightarrow$   (6.1): $\Delta \leftarrow \min_{T \in \mathcal{V}} \delta_T$
8:     **until** $\exists e \notin A : r_e = 0$          (6.2): **for** $e \in E \setminus A$ **do**  $r_e \leftarrow r_e - |\{T : e \in T\}|\Delta$
9:     $A \leftarrow A \cup \{e : r_e = 0\}$
10: **end while**
11: **Output:** $A$

---

$$= [\min_{e \in \mathcal{N}(T)} \frac{r_e^{(t-1)}}{d_e^{(t-1)}}, 0, ..., 0]$$
$$= [\delta_T^{(t)}, 0, ..., 0] \in \mathbb{R}^{n \times 1} \quad \text{(From Line 1 of Algorithm 4)}$$

The next step performs message-passing from the dual node representations $\boldsymbol{h}_T^{(t)}$ and then updates the primal representations $\boldsymbol{h}_e^{(t)}$ via $\boldsymbol{h}_e^{(t)} = g_u(\boldsymbol{h}_e^{(t)}, \sum_{T \in \mathcal{N}(e)} \boldsymbol{h}_T^{(t)})$. We use ELU activation function on the previously computed $\boldsymbol{h}_e^{(t)}$ and a bias $b_r = [1, 0, ..., 0] \in \mathbb{R}^{n \times 1}$. Since $\ln r_e^{(t-1)} \leq 0$, we have:

$$\begin{aligned}
\boldsymbol{h}_e^{(t)} &= \text{ELU}\left(\boldsymbol{h}_e^{(t)}\right) + b_r \\
&= \text{ELU}\left([\ln r_e^{(t-1)}, 0, ..., 0]\right) + [1, 0, ..., 0] \\
&= [\exp(\ln r_e^{(t-1)}) - 1 + 1, 0, ..., 0] \\
&= [r_e^{(t-1)}, 0, ..., 0] \in \mathbb{R}^{n \times 1}
\end{aligned}$$

The update function $g_u$ is also an MLP. We define its weights to be $W_{g_u} = [1, 0, ..., 0, -1, 0, ..., 0]^\top \in \mathbb{R}^{1 \times 2n}$. Then, the hidden dimension $\boldsymbol{h}_e^{(t)}$ for each element $e \in E$ becomes:

$$\begin{aligned}
\boldsymbol{h}_e^{(t)} &= g_u\left(\boldsymbol{h}_e^{(t)}, \sum_{T \in \mathcal{N}(e)} \boldsymbol{h}_T^{(t)}\right) \\
&= W_{g_u}\left[\boldsymbol{h}_e^{(t)} \| \sum_{T \in \mathcal{N}(e)} \boldsymbol{h}_T^{(t)}\right] \\
&= [1, 0, ..., 0, -1, 0, ..., 0]^\top \\
&\quad [r_e^{(t-1)}, 0, ..., 0, \sum_{T \in \mathcal{N}(e)} \delta_T^{(t)}, 0, ..., 0] \\
&= [r_e^{(t-1)} - \sum_{T \in \mathcal{N}(e)} \delta_T^{(t)}, 0, ..., 0] \\
&= [r_e^{(t)}, 0, ..., 0] \in \mathbb{R}^{n \times 1} \quad \text{(From Line 4 of Algorithm 4)}
\end{aligned}$$

Alternatively, if the uniform increase rule is incorporated, we have an additional virtual node $z$ that connects to all dual variables. Its hidden representation $\boldsymbol{h}_z^{(t)}$ is computed as:

$$
\begin{aligned}
\boldsymbol{h}_z^{(t)} &= \min_{T \in \mathcal{T}} \boldsymbol{h}_T^{(t)} \\
&= \min_{T \in \mathcal{T}} [\delta_T^{(t)}, 0, ..., 0] \\
&= [\min_{T \in \mathcal{T}} \delta_T^{(t)}, 0, ..., 0] \\
&= [\Delta^{(t)}, 0, ..., 0] \in \mathbb{R}^{n \times 1} \quad \text{(From Line 6.1 of Algorithm 4)}
\end{aligned}
$$

Then let $\boldsymbol{h}_T'^{(t)} = \boldsymbol{h}_z^{(t)}$, the primal variable updates becomes:

$$
\begin{aligned}
\boldsymbol{h}_e^{(t)} &= g_u \left( \boldsymbol{h}_e^{(t)}, \sum_{T \in \mathcal{N}(e)} \boldsymbol{h}_T'^{(t)} \right) \\
&= W_{g_u} \left[ \boldsymbol{h}_e^{(t)} \| \sum_{T \in \mathcal{N}(e)} \boldsymbol{h}_T'^{(t)} \right] \\
&= [1, 0, ..., 0, -1, 0, ..., 0]^\top \\
&\quad [r_e^{(t-1)}, 0, ..., 0, \sum_{T \in \mathcal{N}(e)} \Delta^{(t)}, 0, ..., 0] \\
&= [r_e^{(t-1)} - d_e^{(t-1)} \Delta^{(t)}, 0, ..., 0] \\
&= [r_e^{(t)}, 0, ..., 0] \in \mathbb{R}^{n \times 1} \quad \text{(From Line 6.2 of Algorithm 4)}
\end{aligned}
$$

For the decoders $q_x$, $q_r$, $q_\delta$ (and $q_\Delta$ if uniform increase is used), they map hidden representations $\boldsymbol{h}_e^{(t)}$, $\boldsymbol{h}_T^{(t)}$ (and $\boldsymbol{h}_z^{(t)}$ if uniform increase is used) to predictions for the intermediate quantities computed by the algorithm. We define $W_{q_x} = W_{q_r} = W_{q_\delta} = W_{q_\Delta} = [1, 0, ..., 0]^\top \in \mathbb{R}^{1 \times n}$. For $x_e^{(t)}$, it is a binary classification task, where $x_e^{(t)} = 1$ if $r_e^{(t)} = 0$ to add the element $e$ to the hitting set. Define $o : \mathbb{R} \to \{0, 1\}$, where

$$
o(x) = \begin{cases} 1, & \text{if } x \le 0 \\ 0, & \text{else} \end{cases}
$$

We note that although we use a sigmoid function in our architecture, the sigmoid function can approximate $o(x)$ to arbitrary precision by adjusting its temperature. Therefore, we have

$$
\begin{aligned}
\hat{x}_e^{(t)} &= o(q_x(\boldsymbol{h}_e^{(t)})) \\
&= o \left( W_{q_x}(\boldsymbol{h}_e^{(t)}) \right) \\
&= o \left( [1, 0, ..., 0]^\top [r_e^{(t)}, 0, ..., 0] \right) \\
&= o \left( r_e^{(t)} \right) \\
&= x_e^{(t)} \quad \text{(Definition of } x_e^{(t)})
\end{aligned}
$$

For the other three intermediate quantities:

$$
\begin{aligned}
\hat{r}_e^{(t)} &= q_r(\boldsymbol{h}_e^{(t)}) \\
&= W_{q_r}(\boldsymbol{h}_e^{(t)}) \\
&= [1, 0, ..., 0]^\top [r_e^{(t)}, 0, ..., 0] \\
&= r_e^{(t)}
\end{aligned}
$$

$$\hat{\delta}_T^{(t)} = q_\delta(\boldsymbol{h}_T^{(t)})$$
$$= W_{q_\delta}(\boldsymbol{h}_T^{(t)})$$
$$= [1, 0, ..., 0]^\top [\delta_T^{(t)}, 0, ..., 0]$$
$$= \delta_T^{(t)}$$
$$\hat{\Delta}^{(t)} = q_\delta(\boldsymbol{h}_z^{(t)})$$
$$= W_{q_\Delta}(\boldsymbol{h}_z^{(t)})$$
$$= [1, 0, ..., 0]^\top [\Delta^{(t)}, 0, ..., 0]$$
$$= \Delta^{(t)}$$

Therefore, $\mathcal{M}_\Theta^{(t)}(\mathcal{T}, E, w) = (\boldsymbol{x}^{(t)}, \boldsymbol{r}^{(t)}, \boldsymbol{\delta}^{(t)}, \Delta^{(t)})$ and the induction step is completed.

## C. Datasets

### C.1. Synthetic datasets

We provide the details of the graph distributions used to generate random graphs for both training and testing.

**Barabási-Albert (B-A) graph** Barabási-Albert graphs for both training and testing are randomly generated using `networkx.barabasi_albert_graph`. The number of edges to attach from a new node to existing nodes is randomly chosen from $[1, 10]$. Node weights for primal variables are uniformly sampled from $[0, 1]$.

**Erdős–Rényi (E-R) graph** We use `networkx.erdos_renyi_graph` to generate random Erdős–Rényi graphs with edge probability uniformly sampled from $[0.2, 0.8]$.

**Star graph** Star graphs are generated by randomly partitioning the nodes into 1 to 5 sets. Within each node set, a star graph is generated with a center node connected to all other nodes. Random edges between the star graphs are then added to ensure the graph is connected.

**Lobster graph** To generate a lobster graph, the number of nodes on the "backbone" $m$ is randomly sampled from $[1, n-1]$, where $n$ is the total number of nodes in the graph. Then, another $k$ nodes are added to the backbone nodes to start "branches", where $1 \le k \le (n - m)$. Finally, the remaining nodes (if any left from $n - m - k$) are randomly attached to the branches.

**3-Connected (3-Con) Planar graph** A 3-regular graph is randomly generated with `networkx.random_regular_graph`, and then checked to see if it is 3-connected and planar. The process is repeated until a valid 3-connected planar graph is found, or if it reaches the limit of 100 attempts.

*Table 5.* Percentage of nodes being in the optimal vertex cover for different graph families.

|          | B-A | E-R | Star | Lobster | 3-Con |
|----------|-----|-----|------|---------|-------|
| 16 nodes | 43% | 71% | 23%  | 40%     | 61%   |
| 32 nodes | 49% | 80% | 15%  | 41%     | 60%   |
| 64 nodes | 45% | 88% | 8%   | 41%     | 59%   |

**Barabási-Albert bipartite graph** We use the same distribution to generate random Barabási-Albert bipartite graph as described in Borodin et al. (2018) and Hayderi et al. (2024). Given parameters $(m, n, b)$, we generate a bipartite graph $G$ on $n$ nodes on the left-hand side (LHS) and $m$ on the right-hand side (RHS) via a preferential attachment scheme. Given $n$ LHS nodes, we attach each RHS node to $b$ LHS nodes sampled without replacement, where the probability of selecting an LHS node $v$ is proportional to $Pr[v] = \frac{\text{degree}(v)}{\sum_{v'} \text{degree}(v')}$.

The parameter $b$ can be varied to generate OOD graphs. For training, we choose $b = 5$. For testing OOD generalization, we use $b = 3$ and $b = 8$ to generate test graphs with sparser and denser preferential attachment.

For vertex cover and set cover, we use Algorithm 2 and Algorithm 3 (Khuller et al., 1994) to generate intermediate supervisions, which are instantiations of Algorithm 1 with improved efficiency, as explained in Section 3.3. For hitting set, we use Algorithm 1 with the uniform increase rule. Furthermore, the optimal solutions are generated with the default IP solver in `scipy`, which is based on HiGHS (Schwendinger & Schumacher, 2023; Huangfu & Hall, 2018).

### C.2. Gurobi datasets

Random B-A graphs are generated following the same distribution described above. For testing, we generate B-A graphs with 500, 600, and 750 nodes. We use the trained model to perform inference on the testing set and retrieve vertex cover solutions. We also use Algorithm 2 (Khuller et al., 1994) to compute solutions with $\epsilon = 0.1$. The comparison of the solutions from the model and the algorithm is shown in Table 6. The two sets of solutions are then used to initialize variables to warm-start the Gurobi solver. We also compare them with Gurobi's default initialization (i.e., no warm start). We use the default parameter settings of Gurobi, setting thread count to 1 (`model.setParam('Threads', 1)`), the time limit to 3600s (`model.setParam('TimeLimit', 3600)`), and random seed (`model.setParam('Seed', seed)`). Each experiment is repeated with 5 seeds.

*Table 6.* The ratio of total weights of the solutions generated by the model compared with those generated by the algorithm ($w_{\text{pred}}/w_{\text{algo}}$). We also report the percentage of uncovered edges from the solutions generated by the model (i.e. how often the cleanup stage is required).

|  | 500 nodes | 600 nodes | 750 nodes |
|---|---|---|---|
| $w_{\text{pred}}/w_{\text{algo}}$ | 0.972 | 0.970 | 0.970 |
| Uncovered edges | 0% | 0% | 0% |

### C.3. Real-world datasets

**Airports** The Airports datasets (Ribeiro et al., 2017) consist of three airport networks from Brazil (131 nodes, 1038 edges), Europe (399 nodes, 5995 edges), and the USA (1190 nodes, 13599 edges). The nodes represent airports, and the edges represent commercial flight routes. The node features are one-hot encoded node identifiers, as described in (Jin et al., 2019). The task is to predict the activity level of each airport, measured by the total number of landings plus takeoffs, or the total number of people arriving plus departing. It is a classification task with 4 labels, with label 1 assigned to the 25% least active airports, and so on, according to the quartiles of the activity distribution. We create 10 random train/val/test splits for the transductive task with a ratio of 60%/20%/20%.

## D. Hyperparameters

All GPU experiments were performed on Nvidia Quadro RTX 8000 with 48GB memory. The Gurobi experiments were conducted on Intel Xeon E7-8890x with 144 cores and 12TB memory.

**Synthetic and Gurobi experiments** For training, we use the Adam optimizer with an initial learning rate of 1e-3 and weight decay of 1e-4, coupled with the ReduceLROnPlateau scheduler with default settings. Additionally, we use a batch size of 32, a hidden dimension of 32, and a maximum of 100 epochs. For testing, we use the trained model with the lowest validation loss.

**Real-world dataset experiments** We use the same optimizer and scheduler settings as in the synthetic experiments. Additionally, we also apply early stopping with a patience of 10 epochs based on validation loss and set the scheduler with a patience of 20 epochs. All embeddings have a fixed dimension of 32. For testing, we use the model with the lowest validation loss. The base models are used with max jumping knowledge (Xu et al., 2018) and L2 normalization after each layer (Rossi et al., 2024). We conduct hyperparameter search for each base model on Airports datasets (Brazil, Europe, USA), then use the setting for all embedding methods. Due to the high computational costs of WikipediaNetwork (Chameleon, Squirrel) and PPI datasets, hyperparameter search is only done with GCN. We use the default TPE hyperparameter search algorithm from `optuna` (Akiba et al., 2019) with a median pruner. The searchable parameters are lr=[0.01, 0.001, 0.0005], hid_dim=[32,

64, 128], dropout=[0.1, 0.3, 0.5], and num_layer=[1, 3, 5]. For training Node2Vec (Grover & Leskovec, 2016), we use walk_length=20, context_size=10, walks_per_node=10, with 100 epochs.

*Table 7.* Additional hyperparameters for real-world dataset experiments.

|  | Brazil | Europe | USA |
|---|---|---|---|
| GCN | lr=0.0005
hid_dim=32
dropout=0.5
num_layer=3 | lr=0.001
hid_dim=32
dropout=0.1
num_layer=3 | lr=0.001
hid_dim=64
dropout=0.3
num_layer=3 |
| GAT | lr=0.001
hid_dim=32
dropout=0.3
num_layer=3 | lr=0.001
hid_dim=32
dropout=0.1
num_layer=3 | lr=0.001
hid_dim=32
dropout=0.1
num_layer=3 |
| SAGE | lr=0.0005
hid_dim=32
dropout=0.5
num_layer=1 | lr=0.0005
hid_dim=32
dropout=0.3
num_layer=1 | lr=0.001
hid_dim=64
dropout=0.1
num_layer=3 |

## E. Additional architectural details

In practice, due to the potentially high degree variance, we apply log transformation on the node degree $d_e^{(t)}$ before encoding it with $f_d$ for better generalization, i.e. $\boldsymbol{h}_{d_e}^{(t)} = f_d(\ln(d_e^{(t-1)} + 1))$. Then, for decoding $\hat{x}_e^{(t)} = q_x(\boldsymbol{h}_e^{(t)})$, which represents whether to include element $e$ to the solution, we apply a sigmoid activation function to convert logits to probabilities. For minimum vertex cover and minimum set cover, since multiple elements can be included into the solution at each timestep, we set a threshold of 0.5 to decide whether to include the element. For minimum hitting set, since the uniform increase rule is used, only one element is included into the solution at each timestep, we choose the element with the highest probability to include in the hitting set. Lastly, we add dropouts in between processor layers with probability of 0.2 for Gurobi and real-world experiments. This helps the model to generalize to much larger graphs at a slight cost of approximation ratios.

## F. Limitation and future work

While our framework is a general one that can learn any algorithm designed with the primal-dual paradigm, the architecture described in the paper can be directly adapted to solve any problem represented by the hitting set formulation. The hitting set provides a flexible structure for modeling various optimization problems by selecting a subset of elements that "hits" or "covers" all required constraints, represented as sets. This formulation is versatile because it can capture diverse constraints (e.g., nodes, edges, paths, cycles), making it applicable to numerous problems. As discussed in Section 3.3, Algorithm 1 can be reformulated to recover many classical (exact or approximation) algorithms for problems that are special cases of the hitting set, covering both polynomial-time solvable and NP-hard problems. Some of these special cases are illustrated in Goemans & Williamson (1996) and Williamson & Shmoys (2011), including shortest s-t path, minimum spanning tree, vertex cover, set cover, minimum-cost arborescence, feedback vertex set, generalized Steiner tree, minimum knapsack, and facility location problems.

We note that not all problems can be directly represented by the hitting set formulation. As a minimization problem, the hitting set does not naturally align with maximization objectives, making the primal-dual approximation algorithm less straightforward to apply. However, the primal-dual framework can still be extended to maximization problems by carefully reformulating the primal-dual pair, where the dual is a minimization problem, and adapting Algorithm 1. Then, similarly, the algorithm starts with a feasible dual solution and iteratively updates both primal and dual variables to reduce the gap between them. Therefore, PDNAR, with its bipartite graph structure, can still be extended to handle maximization problems with appropriate adjustments to Algorithm 1.

Furthermore, while Algorithm 1 provides a general framework for designing primal-dual approximation algorithms, it can be further strengthened by incorporating techniques to enhance the algorithmic performance. One such technique is the uniform increase rule, which we have shown how it can be integrated into our framework. Future work can incorporate other advanced techniques, such as those outlined in (Williamson & Shmoys, 2011), to further extend our framework's ability to

accommodate a broader range of primal-dual approximation algorithms with improved worst-case guarantees.

Lastly, our method may not explicitly preserve the worst-case approximation guarantees of the primal-dual algorithm in practice. Instead, our model focuses on learning solutions that perform better for the training distribution and generalize well to new instances. While this does not mean that the worst-case guarantees are secured, it is important to note that such cases are often less common in real-world scenarios. One of the core strengths of NAR lies in leveraging a pretrained GNN with embedded algorithmic knowledge to tackle real-world datasets. Therefore, we believe it is valuable to train models to produce high-quality solutions for common cases, even if they do not preserve worst-case guarantees. This aligns with the overarching goals of NAR.

## G. Related works on neural algorithmic reasoning

We provide a more comprehensive review of existing works on Neural Algorithmic Reasoning (NAR) and highlight our contributions in this context.

**Neural algorithmic reasoning**    The algorithmic alignment framework proposed by Xu et al. (2020) suggests that GNNs are particularly well-suited for learning dynamic programming algorithms due to their shared aggregate-update mechanism. Additionally, Veličković et al. (2020) demonstrates the effectiveness of GNNs in learning graph algorithms such as BFS and Bellman-Ford. These foundational works have contributed to the development of neural algorithmic reasoning (Veličković & Blundell, 2021), which investigates the potential of neural networks, particularly GNNs, to simulate traditional algorithmic processes. This research direction has since inspired several follow-up studies, including efforts to instantiate the framework for specific algorithms (Deac et al., 2020; Georgiev & Liò, 2020; Zhu et al., 2021), applications to real-world use cases (Deac et al., 2021; He et al., 2022; Beurer-Kellner et al., 2022; Georgiev et al., 2023a; Numeroso et al., 2023; Estermann et al., 2024), architectural improvements (Georgiev et al., 2022; Bevilacqua et al., 2023; Rodionov & Prokhorenkova, 2023; Jain et al., 2023; Engelmayer et al., 2023; Mirjanic et al., 2023; Georgiev et al., 2023b; Dudzik et al., 2024; Jürß et al., 2024; Xhonneux et al., 2024; Georgiev et al., 2024; Rodionov & Prokhorenkova, 2024; Xu & Veličković, 2024; Kujawa et al., 2024), and integration with large language models (Bounsi et al., 2024). Our work advances NAR by introducing a general framework designed to tackle combinatorial optimization problems, particularly NP-hard ones, with the objective of simulating and outperforming primal-dual approximation algorithms.

**Combinatorial optimization with GNNs**    The CLRS benchmark (Veličković et al., 2022) and its extensions (Minder et al., 2023; Markeeva et al., 2024) are widely recognized for evaluating GNNs on 30 algorithms from the CLRS textbook (Cormen et al., 2001) and more. However, these algorithms are limited to polynomial-time problems, leaving the more challenging NP-hard problems largely unexplored in neural algorithmic reasoning. A comprehensive review by Cappart et al. (2022) summarizes the current progress of using GNNs for combinatorial optimization. The most relevant work (Georgiev et al., 2023c) trains GNNs on algorithms for polynomial-time-solvable problems and test them on NP-hard problems, demonstrating the value of algorithmic knowledge over non-algorithmically informed models. In contrast, our approach bridges this gap by extending GNNs to tackle NP-hard problems through the use of primal-dual approximation algorithms. Furthermore, we integrate optimal solutions from integer programming, which guides the model toward better outcomes during training. To the best of our knowledge, our method is the first of its kind to surpass the performance of the algorithms it was originally trained on.

**Multi-task learning for NAR**    Early work by Veličković et al. (2020) demonstrated that BFS and Bellman-Ford are best learned jointly, and subsequent studies have highlighted broader benefits of multi-task learning when GNNs are trained on multiple algorithms simultaneously (Xhonneux et al., 2021; Ibarz et al., 2022). Building on this, Numeroso et al. (2023) leveraged the primal-dual principle from linear programming to successfully learn the Ford-Fulkerson algorithm using the max-flow min-cut theorem. However, their approach was tailored specifically for Ford-Fulkerson and did not generalize to other primal-dual scenarios or address NP-hard problems. To overcome these limitations, our work introduces a general framework that employs the primal-dual principle to enable GNNs to benefit from multi-task learning across a broad range of optimization problems, particularly those expressible as instances of the general hitting set problem.

## H. Contribution to Neural Combinatorial Optimization (NCO)

Most GNN-based supervised learning methods for NCO learn task-specific heuristics or optimize solutions in an end-to-end manner (Joshi et al., 2019; Li et al., 2018; Gasse et al., 2019; Fu et al., 2021). These end-to-end approaches rely exclusively on supervision signals derived from optimal solutions, which are computationally expensive to obtain for hard instances. Furthermore, dependence on such labels can limit generalization (Joshi et al., 2022). In contrast, our method trains on synthetic data obtained efficiently from a polynomial-time approximation algorithm. The embedding of algorithmic knowledge also demonstrates strong generalization. The addition of optimal solutions are derived from small problem instances, enabling our model to outperform the approximation algorithm and generalize effectively to larger problem sizes.

Our approach represents a previously underexplored area of NCO research with GNNs, offering a general framework to build an algorithmically informed GNN to tackle combinatorial optimization problems. Unlike end-to-end methods, we leverage intermediate supervision signals from polynomial-time approximation algorithms, which can be generated efficiently, to address key bottlenecks in data efficiency and generalization. Additionally, we fill the gap in autoregressive methods for NCO using GNNs by aligning our architecture with the primal-dual method, enabling the GNN to simulate a single algorithmic step in an efficient and structured manner.

We provide additional empirical evidences on the RB benchmark graphs, which are well-known hard instances for the Minimum Vertex Cover (MVC) problem. We compared our method with several NCO baselines. Given the supervised nature of our approach, obtaining optimal solutions for RB200/500 graphs is computationally challenging. Therefore, we tested the generalization of our model, trained on Barabási-Albert graphs of size 16 (using intermediate supervision from the primal-dual algorithm and optimal solutions), on the larger RB200/500 graphs. We compare with EGN (Karalias & Loukas, 2021) and Meta-EGN (Wang & Li, 2023), two powerful NCO baselines for these benchmarks, as well as two algorithms (the primal-dual approximation algorithm and the greedy algorithm) and Gurobi. The results are summarized in Table 8.

*Table 8.* Approximation ratio of solutions compared with optimal solutions for MVC (lower is better).

| Method | RB200 | RB500 |
|---|---|---|
| EGN | $1.031 \pm 0.004$ | $1.021 \pm 0.002$ |
| Meta-EGN | $1.028 \pm 0.005$ | $1.016 \pm 0.002$ |
| PDNAR (ours) | $1.029 \pm 0.005$ | $1.020 \pm 0.004$ |
| Primal-dual | $1.058 \pm 0.003$ | $1.056 \pm 0.004$ |
| Greedy | $1.124 \pm 0.002$ | $1.062 \pm 0.005$ |
| Gurobi 9.5 ($\leq 1.00$s) | $1.011 \pm 0.003$ | $1.019 \pm 0.003$ |
| Gurobi 9.5 ($\leq 2.00$s) | $1.008 \pm 0.002$ | $1.019 \pm 0.003$ |

From Table 8, we observe that PDNAR outperforms EGN and is competitive with its improved variant, Meta-EGN. Notably, EGN and Meta-EGN are trained directly on 4000 RB200/500 graphs, while our method was trained solely on 1000 Barabási-Albert graphs with just 16 nodes and tested out-of-distribution. This highlights the data efficiency and strong generalization capability of PDNAR.

We note that both EGN and Meta-EGN are unsupervised methods, whereas our method adopts a supervised approach. A fairer comparison would involve supervised baselines for NCO. However, current focus of NCO has primarily shifted towards unsupervised methods, leaving existing supervised baselines with outdated performances. Obtaining labels for large graphs, such as RB200/500, is also computationally prohibitive. Additionally, supervised methods usually require a combination of external solvers or search algorithms, which is a different setting than ours. In summary, our supervised approach addresses this underexplored direction of NCO by addressing its key challenges — PDNAR as an algorithmically informed GNN that leverages efficiently obtainable labels while demonstrating strong generalization capabilities.

## I. Duality in linear programming

Duality in linear programming has been utilized in training neural networks to tackle optimization problems. Li et al. (2024) introduces a Learning-to-Optimize method to mimic Primal-Dual Hybrid Gradient method for solving large-scale LPs. While they focus on developing efficient solvers for LPs, we aim to simulate the primal-dual approximation algorithm for NP-hard problems using GNNs. Although both approaches reference the primal-dual framework, this similarity is superficial.

The primal-dual terminology is widely used in optimization, but our work applies it to study algorithmic reasoning. For example, the primal-dual approximation algorithm can be instantiated to many traditional algorithms, such as Kruskal's algorithm for MST. Furthermore, unlike Li et al. (2024), our method relies on intermediate supervision from the primal-dual algorithm to guide reasoning, ensuring that the model learns to mimic algorithmic steps. We also incorporate optimal solutions into the training process to improve solution quality, allowing our model to outperform the primal-dual algorithm it is trained on. Moreover, the architectures differ significantly: our method employs a recurrent application of a GNN to iteratively solve problems, while Li et al. (2024) does not use GNNs or recurrent modeling. These distinctions highlight that our focus is not on solving LPs but on leveraging NAR to generalize algorithmic reasoning for NP-hard problems.

