# OpenReview forum: "Primal-Dual Neural Algorithmic Reasoning"
_ICML.cc/2025/Conference — ICML 2025 spotlightposter_

### Official Review · Reviewer_aupK · 2025-03-07

**Overall Recommendation:** 4

**Summary:**

This work proposed a framework PDNAR, that lies within the neural algorithmic reasoning framework, uses a bipartite MPNN to simulate the primal dual algorithm on solving minimum hitting set problem and its extensions.

**Claims And Evidence:**

They provided theoretical proof that MPNN can simulate the algorithm.
They provided experimental results that PDNAR outperforms the baselines in all the experiments.

**Essential References Not Discussed:**

Not that I know, they discussed wide related work in the appendix as well.

**Experimental Designs Or Analyses:**

I checked all the experiments.

**Methods And Evaluation Criteria:**

The problem definition is solid, the proposed method makes sense and the benchmark datasets (both real world and synthetic) are good.

**Other Comments Or Suggestions:**

Some notations are messy, for example some symbols $E, A$ are overridden without clarifying.

**Other Strengths And Weaknesses:**

Strength:
- Overall the paper is well written and easy to follow.
- The novelty is significant, and the experiment results are pretty strong.

Weakness:
- The algorithm 1 is hard to read without explanation.

**Questions For Authors:**

- You remove the variables in the bipartite graph by masking them out, but how? By assigning 0s? Does it influence batchnorm or something?
- Could you explain more what is the NAR baseline and how is it essentially different from PDNAR? Because PDNAR is also NAR.

**Relation To Broader Scientific Literature:**

The work focused on MHS problem and its extensions, which are some essential families of CO problems.
The work also established the effectiveness of bipartite representation and MPNN approach for the primal dual algorithm.

**Theoretical Claims:**

I check the proof in the appendix.
The assumption of MPNN is that it uses MLPs. But there exist some functions like ln or fractional that cannot directly be simulated by MLPs.

---

> ### Author Rebuttal · Authors · 2025-04-01
>
> Thank you for your support and highlighting the strengths of our paper, which we summarize below.
> - Paper: **well-written** and **easy to follow**
> - Problem definition: **solid**
> - Novelty:  **significant**
> - Benchmark datasets: **good**
> - Empirical results: **strong**
>
> We address your questions in the following:
>
> > I check the proof in the appendix. The assumption of MPNN is that it uses MLPs. But there exist some functions like ln or fractional that cannot directly be simulated by MLPs.
>
> Thanks for checking our proof in the appendix! As shown in Line 767, the logarithm is hard-coded as a feature transformation, and thus is not to be simulated by MLPs. The logarithm is used so that the fractional/division can be achieved by subtraction followed by the ELU activation function. This avoids simulating division with MLPs.
>
> > You remove the variables in the bipartite graph by masking them out, but how? By assigning 0s? Does it influence batchnorm or something?
>
> When removing a variable, we remove all its edges in the bipartite graph, so it does not effectively participate in future message-passing steps. The edges are removed by applying a true/false mask. Empirically, we do not apply BatchNorm.
>
> > Could you explain more what is the NAR baseline and how is it essentially different from PDNAR? Because PDNAR is also NAR.
>
> The processor of PDNAR was specially designed to align with the primal-dual approximation algorithm (Section 4.1). For the NAR baseline, we use the conventional choice of MPNN with max aggregation (e.g. [1]). Furthermore, the NAR baseline does not use the primal-dual bipartite formulation, and therefore, is only trained on the primal but not the dual signal. Results align with the previous findings that multi-task learning helps in NAR (e.g. [1]). We will edit “Baselines” in Section 5.1 to highlight the differences and architectural details for the NAR baseline.
>
> > The algorithm 1 is hard to read without explanation. Some notations are messy.
>
> Thank you for pointing out the areas that we can improve for the readability! We will enhance the explanation of the algorithm in Section 3.3 and make the pseudocode more intuitive to understand. We will also include a notation table and ensure the notational consistency in our revised paper.
>
> We thank the reviewer for your valuable feedback. We hope these answer your questions, and please let us know if you require further clarification. Thank you very much!
>
>
> [1] Neural Execution of Graph Algorithms, Veličković et al, ICLR 2020.

---

### Official Review · Reviewer_WLAv · 2025-03-10

**Overall Recommendation:** 5

**Summary:**

The authors present a neural architecture adopting the primal-dual framework, as studied in algorithm design especially for the approximation of NP-hard problems.  Using the minimum hitting set as the primary case study, the authors prove the proposed architecture satisfies the requirement of algorithmic alignment, i.e., it can replicate the intermediate states produced by the actual algorithm as described in Algorithm 1.  Importantly, the authors propose a training strategy enabling the neural architecture to learn from small instances with known solutions.  Beyond replicating the algorithm, the resulting neural solver can surpass the quality of the solvers used to train it, and the embeddings computed by the trained neural architecture can have other benefits.  A comprehensive suit of experiments adequately support those claims, establishing the efficacy of the proposed approach.  Further valuable discussion is presented in the appendices, e.g., more discussion of related work, as well as a summary of limitations.

**Claims And Evidence:**

- A key claim is algorithmic alignment, which is supported by a theoretical proof in Appendix B.
- Empirical results further demonstrate the efficacy of the approach and proposed architecture.

**Essential References Not Discussed:**

N/A

**Experimental Designs Or Analyses:**

Each section seemed to use reasonable datasets, e.g., known graph families, datasets used in prior studies, or comparing to established solvers.

**Methods And Evaluation Criteria:**

The contributions are supported by a comprehensive set of evaluations of synthetic, OOD, and real-world datasets, along with a comparison to commercial solvers.

**Other Comments Or Suggestions:**

I see Appendix F mentions possible strengthening using ``other advanced techniques'' without specifics.  It would have helped to even hint at 1-2 such techniques.

**Other Strengths And Weaknesses:**

I was hoping the authors would do an ablation study of the uniform update rule.

**Questions For Authors:**

No further questions at this time.

**Relation To Broader Scientific Literature:**

The authors adequately discuss related works in algorithm theory, neural reasoning, neural combinatorics optimization, and linear programming.

**Theoretical Claims:**

Only glanced at the proofs in Appendix B.  The presented strong induction seems to make sense.  It's a bit difficult to follow given its length, mirroring the steps of the algorithm.  It would help to recall the pseudocode, highlighting which part is being mirrored at each proof step.

One remark: the last line mentioned $\Theta_{\text{dual}}$ while I expected $\mathcal{M}_\Theta$.

---

> ### Author Rebuttal · Authors · 2025-04-01
>
> We sincerely thank the reviewer for your time carefully **reading our main paper and appendices** and for **strongly supporting** our paper! Thank you for highlighting our experiments and discussion in the appendix as comprehensive and helpful.
>
> We answer your questions in the following:
>
> > I was hoping the authors would do an ablation study of the uniform update rule.
>
> We conducted an ablation by comparing the model with and without the uniform update rule on the MHS (minimum hitting set) problem. Recall that the primal-dual approximation algorithm for MHS requires the uniform update rule. Results in Table R1 show that the uniform update rule design in our model helps to improve the model performance due to its better alignment with the algorithm.
>
> Table R1. Ablation of the uniform update rule on MHS.
>
> | Uniform update rule | 16 (1x)           | 32 (2x)           | 64 (4x)           | 128 (8x)          | 256 (16x)         | 512 (32x)         | 1024 (64x)        |
> |----------------------|-------------------|-------------------|-------------------|-------------------|-------------------|-------------------|-------------------|
> | No                   | 0.996 ± 0.003     | 0.983 ± 0.010     | 0.999 ± 0.013     | 0.987 ± 0.013     | 0.977 ± 0.005     | 1.009 ± 0.013     | 1.060 ± 0.028     |
> | Yes                  | 0.990 ± 0.003     | 0.981 ± 0.003     | 0.983 ± 0.006     | 0.965 ± 0.005     | 0.979 ± 0.005     | 1.004 ± 0.013     | 1.043 ± 0.022     |
>
>
>
>
> > I see Appendix F mentions possible strengthening using ``other advanced techniques'' without specifics. It would have helped to even hint at 1-2 such techniques.
>
>
> Yes, similar to the uniform update rule, there are other advanced techniques to enhance the basic primal-dual framework to design approximation algorithms. One example is selectively choosing the number of dual variables to increase at each timestep, strengthening scalability. Another example is specialized handling of dual variables with different types, which is required for the uncapacitated facility location problem. These techniques can be incorporated into our framework as extensions, which we leave for future work. We will add more specifics and examples about these techniques in our revised paper.
>
> > Only glanced at the proofs in Appendix B... It would help to recall the pseudocode, highlighting which part is being mirrored at each proof step.
>
> Thanks for reading our proof and providing helpful suggestions on its readability! We will include a side-by-side reference to the algorithm’s pseudocode in Appendix B. And thank you for pointing out the typo. We will fix it in our revision as well.
>
>
> We are grateful for your support and hope we have addressed your questions. Please let us know if you require further clarification. Thank you very much!

---

### Official Review · Reviewer_jk4v · 2025-03-13

**Overall Recommendation:** 3

**Summary:**

This paper presents a general NAR framework based on the primal-dual paradigm, aiming to solve NP-hard problems that traditional NAR methods struggle with by mimicking approximation algorithms. The authors provide a detailed model description, theoretical justifications, and empirical validation on three NP-hard graph tasks. Overall, I find the paper novel and well-written. However, I still have concerns regarding certain details. If the authors can address these issues, I would be willing to reconsider my score.

**Claims And Evidence:**

yes

**Essential References Not Discussed:**

yes

**Experimental Designs Or Analyses:**

yes

**Methods And Evaluation Criteria:**

yes

**Other Comments Or Suggestions:**

see weakness

**Other Strengths And Weaknesses:**

Strengths:
1. This paper extends traditional NAR to NP-hard problems, which I find meaningful and valuable.
2. The paper provides solid theoretical support for its approach.
3. The writing is clear, and the paper is well-structured and well-organized.


Weaknesses:

1. The proposed method aims to directly learn approximation algorithms for NP-hard tasks. However, the authors only discuss three NP-hard problems related to graphs, which may limit the generality of their approach.

2. I am curious about how this method performs on problems that are not NP-hard. If it can effectively approximate both NP and non-NP problems, it would make the approach even more compelling.

3. For NP problems, should the authors consider multiple approximation algorithms rather than a single one?

4. I remain concerned about the construction of effective training and testing sets, as obtaining reliable ground truth for NP-hard problems is inherently challenging.

**Questions For Authors:**

see weakness

**Relation To Broader Scientific Literature:**

This paper extends the traditional NAR to NP-hard problems

**Theoretical Claims:**

yes

---

> ### Author Rebuttal · Authors · 2025-04-01
>
> Thank you for your positive feedback on our paper, which we summarize below.
> - Paper: **novel** and **well-written**
> - Motivation: **meaningful** and **valuable**
> - Theoretical support: **solid**
> - Writing: **clear**, **well-structured and well-organized**
>
> We address your questions in the following:
>
> > The proposed method aims to directly learn approximation algorithms for NP-hard tasks. The authors only discuss three NP-hard problems related to graphs, which may limit the generality of their approach.
>
> We politely point out that while Minimum Vertex Cover is a graph problem, both Minimum Set Cover and Minimum Hitting Set are not graph problems. Furthermore, Minimum Hitting Set is a general formulation of a wide range of graph and non-graph problems. What we propose was a bipartite graph representation which “turns” these problems into graphs using primal and dual variables. This is based on the fact that all linear programs have a dual formulation.
>
>
>
> > I am curious about how this method performs on problems that are not NP-hard. If it can effectively approximate both NP and non-NP problems, it would make the approach even more compelling.
>
> Thank you for highlighting that our primal-dual framework applies to both NP-hard and polynomial-time solvable problems. While this is true, we emphasize that our main contribution lies in the NP-hard domain. In NAR, NP-hard tasks were previously underexplored due to the lack of optimal solutions and the complexity of the problems. Therefore, we focus on extending NAR beyond exact algorithms for tractable problems and into the space of approximation algorithms for NP-hard tasks. Furthermore, the goals are different. In tractable problems, the aim is to simulate an algorithm that *already* produces optimal solutions. In our case, the objective is to outperform the approximation algorithm itself. To do this, we leverage optimal solutions from integer programming solvers, which is only meaningful in the NP-hard setting. As a result, we evaluate performance based on solution quality (weight ratio), rather than accuracy as commonly used in prior NAR work on tractable problems. We elaborate on this point in the next response.
>
>
> > I remain concerned about the construction of effective training and testing sets, as obtaining reliable ground truth for NP-hard problems is inherently challenging.
>
> We clarify that only *training* requires obtaining ground-truth labels. Since we train on small problem instances on the scale of 16, these are fast to obtain with integer programming solvers like HiGHS and Gurobi. Then, we *test* against the approximation algorithm on larger problems. As shown in the caption of Table 1, the metric we use is the *model-to-algorithm weight ratio*: the sum of weights from the model solution divided by the sum of weights from the algorithm solution ($w_{model}/w_{algo}$). Therefore, a weight ratio < 1 means the model produces higher-quality solutions than the algorithm. In summary, we show that training on these small optimal instances enables the model to outperform the algorithm on larger problems (up to scale of 1024).
>
>
> > For NP problems, should the authors consider multiple approximation algorithms rather than a single one?
>
> Thank you for suggesting this! Yes, training an NAR model on multiple algorithms simultaneously has been proven to be effective [1, 2]. Therefore, we hypothesize that our models can also enjoy multi-task learning benefits by training on multiple approximation algorithms simultaneously. However, the multi-task training setup requires more intricate designs, such as the selection of approximation algorithm pairs/sets and architectural design. We believe this is outside the scope of the current paper, but it points to a meaningful direction for future work: how multi-task learning transfers from exact algorithms to approximation algorithms for NAR. We will add these points to Future Work in the revised paper.
>
> We sincerely thank the reviewer for your valuable feedback and highlighting interesting future directions. We hope we answered your questions, and please don’t hesitate if you require further clarification. Thank you very much!
>
>
> [1] A Generalist Neural Algorithmic Learner, Ibarz et al, LoG 2022.
> [2] How to transfer algorithmic reasoning knowledge to learn new algorithms? Xhonneux et al., NeurIPS 2021.

---

### Official Review · Reviewer_MtfB · 2025-03-16

**Overall Recommendation:** 4

**Summary:**

The authors propose Primal-Dual Neural Algorithmic Reasoning (PDNAR),  for training neural networks to simulate classical approximation algorithms. The core idea is to leverage primal-dual paradigm,  by representing primal and dual variables as a bipartite graph and parameterzing by GNN. Optimal solutions from small problem instances are incorporated as training signals, enabling the network to surpass the performance of the original algorithms.

The authors demonstrate that their method outperforms existing baselines acrosss NP-hard problems like vertex cover, set cover, and hitting set.

Code is also shared.

**Claims And Evidence:**

1. Generalization to larger graphs: Results shown in Table 1. Trained on smaller, inference on larger.. better quality.
2. Generalize to graphs from OOD families: Table 2. OOD samples were used for evaluation.
3. Application of warm start large-scale commercial solvers, such as Gurobi: Table 4, better initialization leads to lower running time,

**Essential References Not Discussed:**

NA

**Experimental Designs Or Analyses:**

Yes. The evaluation seems correct.

**Methods And Evaluation Criteria:**

Yes.

**Other Comments Or Suggestions:**

Check questions for authors.

**Other Strengths And Weaknesses:**

Strengths

1. Ablations are done with and without optimal solution integration. (  no optm).. Without optm the quality reduces.
2. Application is shown -> Initialization  Gurobi helps in reducing running time.
3. Code is shared.

**Questions For Authors:**

1. Are there any failure cases? What kind of scenarios should use this approach? Where does it fail?
2. What if optimal solution is not easy to obtain? Wow does the method fare when high quality but non-optimal solutions are available). I believe if this helps then its an advantage in cases where obtain optimal for small instances is also hard. are there any such cases? How would the algorithm fare. This is an optional expt. But a small discussion(even without experiments) on this should improve the paper.

**Relation To Broader Scientific Literature:**

PDNAR directly trains GNNs to simulate and improve upon approximation algorithms for NP-hard tasks.
PDNAR goes beyond simply replicating algorithms by incorporating optimal solutions from small problem instances as training signals. This helps them outperform existing method.

**Theoretical Claims:**

Not checked in detail.

---

> ### Author Rebuttal · Authors · 2025-03-31
>
> Thank you for your time reading our paper and providing valuable feedback!
>
> We appreciate your **positive** acknowledgement of our **empirical design**, **results**, and **application**.
>
> We address your questions in the following:
>
> > Are there any failure cases? What kind of scenarios should use this approach? Where does it fail?
>
> We note that the primal-dual framework is a core method for many algorithmic problems, so our proposed framework is general. On the other hand, our current architecture design is based on the hitting set. While hitting set can be reformulated as many other problems (e.g. vertex cover, set cover, etc), problems that are not directly formulated as the hitting set may require further adjustment. For example, the uncapacitated facility location problem has two types of dual variables that require different handling, which our current model does not adapt to. We believe these can be interesting future directions to extend our model to an even broader set of algorithms. We will add more elaboration and examples in the revised paper.
>
> > What if optimal solution is not easy to obtain? Are there any such cases? How would the algorithm fare.
>
> We only train on very small instances on the scale of 16, so obtaining optimal solutions is not a problem, especially with advanced integer programming solvers like HiGHS and Gurobi. Futhermore, while optimal solutions give an advantage to our model to surpass the approximation algorithm, we included an ablation to train with the algorithmic steps only. This setup corresponds to “No optm” in Table 1. We can see in MVC, “No optm” outperforms traditional NAR, and has better generalization ability than both NAR and TripletMPNN (the latter is much more computationally expensive). This highlights the strength of our model, which aligns closely with the primal-dual framework and leverages both primal and dual training signals. We will add a discussion of these points in Section 5.1.
>
> Thank you so much for the questions! We hope these answers address them. Please let us know if you require further clarification!

---

> > ### Comment · Reviewer_MtfB · 2025-04-05
> >
> > Thanks a lot.
> > I increase my score.

---

> > > ### Author Response · Authors · 2025-04-05
> > >
> > > Thank you so much for reading our rebuttal and increasing the score!

---

### Decision · Program_Chairs · 2025-05-01

**Decision:**

Accept (spotlight poster)

**Comment:**

The paper proposes to train neural networks that can simulate classical approximative algorithms for optimization problems. The underlying GNNs can simulate primal-dual algorithms. Experimentally, their method is able to outperform some baselines on a few classical combinatorial problems.
Reviewers note good experimental results, novelty and being well-written and all recommend acceptance. The rebuttal is also positively acknowledged. Therefore the final recommendation is also to acept the paper.